# The evolution and international spread of extensively drug resistant *Shigella sonnei*

Lewis C. E. Mason [1,2], David R. Greig[3], Lauren A. Cowley[4], Sally R. Partridge [5,6,7,8], Elena Martinez[7,9], Grace A. Blackwell[7,9], Charlotte E. Chong[2], P. Malaka De Silva[2], Rebecca J. Bengtsson[2], Jenny L. Draper [7,9], Andrew N. Ginn[7,8,9,10], Indy Sandaradura [6,7,9], Eby M. Sim[5,6,8], Jonathan R. Iredell[5,6,7,8], Vitali Sintchenko [5,6,7,8,9,11], Danielle J. Ingle [12], Benjamin P. Howden [12], Sophie Lefèvre [13], Elisabeth Njamkepo [13], François-Xavier Weill [13], Pieter-Jan Ceyssens[14], Claire Jenkins[3] & Kate S. Baker [1,2] ✉

*Shigella sonnei* causes shigellosis, a severe gastrointestinal illness that is sexually transmissible among men who have sex with men (MSM). Multidrug resistance in *S. sonnei* is common including against World Health Organisation recommended treatment options, azithromycin, and ciprofloxacin. Recently, an MSM-associated outbreak of extended-spectrum β-lactamase producing, extensively drug resistant *S. sonnei* was reported in the United Kingdom. Here, we aimed to identify the genetic basis, evolutionary history, and international dissemination of the outbreak strain. Our genomic epidemiological analyses of 3,304 isolates from the United Kingdom, Australia, Belgium, France, and the United States of America revealed an internationally connected outbreak with a most recent common ancestor in 2018 carrying a low-fitness cost resistance plasmid, previously observed in travel associated sublineages of *S. flexneri*. Our results highlight the persistent threat of horizontally transmitted antimicrobial resistance and the value of continuing to work towards early and open international sharing of genomic surveillance data.

Bacteria belonging to the genus *Shigella* cause shigellosis, a gastro-intestinal infection characterised by a combination of profuse, often bloody, diarrhoea with abdominal pain, fever, nausea and/or vomiting[1]. In middle- and higher-income nations, although symptoms can be severe, the duration of illness is usually short-lived[2]. However, in lower to middle income countries (LMIC), shigellosis is associated with higher levels of morbidity and mortality[3]. *Shigella* infection has no substantial non-human reservoir and is transmitted from person to

[1]NIHR HPRU in Gastrointestinal Infections at University of Liverpool, Liverpool, UK. [2]Department of Clinical Infection, Microbiology, and Immunology; Institute for Infection, Veterinary and Ecological Sciences, Liverpool, UK. [3]Gastro and Food Safety (One Health) Division, UK Health Security Agency, London, UK. [4]Milner Centre for Evolution, University of Bath, Bath, UK. [5]Centre for Infectious Diseases and Microbiology, The Westmead Institute for Medical Research, Westmead, NSW, Australia. [6]Western Sydney Local Health District, Westmead, NSW, Australia. [7]Faculty of Medicine and Health, University of Sydney, Sydney, NSW, Australia. [8] Sydney Infectious Diseases Institute, University of Sydney, Sydney, NSW, Australia. [9]New South Wales Health Pathology, Dee Why, NSW, Australia. [10]Douglass Hanly Moir Pathology, Macquarie Park, NSW, Australia. [11]Centre for Infectious Diseases and Microbiology – Public Health, Institute for Clinical Pathology and Microbiology Research, Westmead Hospital, Westmead, NSW, Australia. [12]Department of Microbiology and Immunology, The University of Melbourne at The Peter Doherty Institute for Infection and Immunity, Melbourne, Australia. [13]Institut Pasteur, Université Paris Cité, Unité des Bactéries pathogènes entériques, Centre National de Référence des Escherichia coli, Shigella et Salmonella, Paris F-75015, France. [14]Division of Human Bacterial Diseases, Sciensano, Ixelles, Belgium. ✉e-mail: kbaker@liverpool.ac.uk

person by contact with an infected individual[1], or by the consumption of food and/or water contaminated by human faeces[4,5]. The four species of *Shigella* (*S. sonnei*, *S. flexneri*, *S. boydii* and *S. dysenteriae*) are responsible for approximately 212,400 deaths per year globally, 30% of which occur in young children aged under five years old[3]. In the UK, approximately 1650 cases of shigellosis are notified to UK Health Security Agency (UKHSA) each year, and *S. sonnei* is the most frequently detected species[6]. Prior to the 1980s, outbreaks of *S. sonnei* in the UK and other higher-income nations were associated with schools and other institutional settings[7]. However, in recent decades more infections were imported by travellers returning from low and middle income countries (LMIC)[8], sometimes followed by on-going transmission within close knit communities[9]. Since 2012, sexual transmission of *S. sonnei* between men who have sex with men (MSM) has contributed to a high proportion of diagnoses in the UK[10].

*S. sonnei* already exhibited multidrug resistance (MDR), to sulphonamides, ampicillin, streptomycin, and tetracycline, throughout the 1960s, conferred by different combinations of antimicrobial resistance (AMR) genes[11,12]. As a single serotype species, the splitting of *S. sonnei* into four genomic lineages[13], and further clades and subclades[14], has facilitated the study of its relationship with AMR and surveillance of global transmission. For example, a study of the introduction of *S. sonnei* into Vietnam highlighted the ability of *Shigella* to refine its accessory genome repertoire whereby introduced subclades were observed to acquire AMR genes, and genes encoding colicins that facilitate killing of other bacteria[15]. These characteristics contributed to its ability to become endemic in the human population and evolve over time towards resistance to an increasing number of antimicrobials. Furthermore, a recently defined subclade of the globally dominant genomic subtype of *S. sonnei*, Lineage 3[13], disseminating from South-East Asia, has acquired ciprofloxacin resistance, conferred by three mutations in the quinolone resistance determining region (QRDR) of the *gyrA* and *parC* genes[16]. More recently, ciprofloxacin resistant *S. sonnei* has acquired resistance to azithromycin, one of the second line agents, particularly in communities where antibiotic use is high[17]. The first global transmission of azithromycin resistant shigellosis was described among MSM, driven by the acquisition of a low-fitness cost AMR plasmid, and frequent use of antimicrobials for other sexually transmitted illnesses (STIs) leading to bystander resistance[18,19]. The acquisition of resistance to ciprofloxacin, recommended by the World Health Organisation (WHO) for empirical treatment of shigellosis[20], and increasing MDR in *S. sonnei* has earned it a place as a WHO priority organism for which new antimicrobials are urgently needed[21].

The extensive global dissemination of MDR *S. sonnei* subsequently led to periodic outbreaks of XDR *S. sonnei*, defined here as being resistant to the only first-line antimicrobial (ciprofloxacin) and to all but one of the second-line antimicrobials (pivmecillinam, ceftriaxone and azithromycin) recommended by the WHO for the treatment of shigellosis[20]. These outbreak isolates have resistance to ciprofloxacin, azithromycin and extended-spectrum β-lactam antibiotics (e.g., ceftriaxone)[22]. The latter resistance is typically associated with acquisition of an extended-spectrum β-lactamase (ESBL) gene, such as those in the $bla_{CTX-M}$ family. Sporadic, $bla_{CTX-M}$-containing XDR *S. sonnei* isolates and local outbreaks have been previously described in various countries[14,23], but there has been little evidence of widespread dissemination of these subtypes, unlike their ciprofloxacin and ciprofloxacin-azithromycin resistant predecessors. Since 2015, transmission of MSM-associated shigellosis in the UK has been dominated by *S. sonnei* resistant to both azithromycin and ciprofloxacin[6], until the COVID-19 pandemic led to a sharp reduction in case numbers[24]. However, during the resurgence of shigellosis in the last quarter of 2021, an outbreak of *S. sonnei* occurred that likely involved multiple countries[22,25,26]. To further explore the strain responsible for the newly sustained domestic transmission of XDR *S. sonnei* in the UK and explore its potential links to strains internationally (only possible with

genomics in this single serotype pathogen), we use genomic epidemiology to study the evolution of this strain from its MDR predecessor, the genetic basis for the XDR genotype, and the extent of dissemination across Europe, Australia, and the United States of America (USA).

## Results
### Evolutionary history of the UK XDR outbreak
To explore the evolutionary history and global context of *S. sonnei* belonging to CipR.MSM5 and specifically those isolates associated with the XDR outbreak in the United Kingdom, we constructed a cgMLST tree of a subset of 2820 clinical isolates of *S. sonnei* processed by UKHSA between 2016 and 2021, alongside 120 subclade representatives of the global genotyping framework (Fig. 1)[14]. The majority (*n* = 2730/2820, 97%) of UK isolates belonged to the globally dominant *S. sonnei* Lineage 3. Among these isolates, overlaying the year of isolation revealed a pattern of clonal replacement whereby isolates collected in later years tended to cluster together in more distant parts of the phylogeny and the sustained domestic transmission of a ciprofloxacin resistant sublineage (Fig. 1, Supplementary Fig. 1). This is consistent with previous reports of CipR.MSM5 (otherwise known as genotype 3.6.1.1.2) replacing earlier circulating sublineages of *S. sonnei* in the UK[6,17]. Among the UK isolates, predicted ESBL activity was solely associated with the presence of a $bla_{CTX-M}$ genes (*n* = 567, Supplementary Data 1). Historically, these genes had been seen only sporadically in conjunction with *mph*(A), conferring azithromycin resistance in the ciprofloxacin resistant lineage (Fig. 1). More recently however, a sub-clade (a 10 SNP single linkage cluster, called t10.377) acquired $bla_{CTX-M-27}$, to give rise to the XDR outbreak cluster (approximated by BAPS5 and indicated by the presence of $bla_{CTX-M-27}$ at the extremity of the tree) such that 90% (*n* = 35/39) of *S. sonnei* isolates from 2021 belonged to the XDR-outbreak (Fig. 1, see Supplementary Fig. 1 for full genotyping).

The patient demography of isolates in the outbreak linkage cluster supported epidemiological investigations that confirmed this was an MSM-associated outbreak[26]. Specifically, isolates from male patients aged 16 to 60 years who had not reported recent international travel were overrepresented among the outbreak isolates, with these patients contributing 84% (405/482) of the UK isolates in t10.377 compared with 20% (475/2412) of UK isolates in the remainder of the phylogeny (*p*-value < 0.0001). We also found topological clustering of travel-associated isolates, i.e., isolates from patients recently returned from Asia co-located in the phylogeny, as did those from Africa (Fig. 1), reflecting the different bacterial populations endemic to these regions; a previously-described phenomenon[17,18,27]. The MSM-associated outbreak lineage (t10.377) was almost exclusively (99%, 478/482) comprised of isolates from patients who did not report travel to Asia, Africa, or Latin America, compared to 69% (1661/2412) in the rest of the UK phylogeny (*p*-value < 0.00001).

### International linkage and accessory genome evolution of the outbreak
We explored potential international links and obtained a more detailed evolutionary history of the UK outbreak cluster (t10.377) by integrating and analysing datasets from international partners (Fig. 2). The subset of UK isolates belonging to t10.377 (*n* = 468) were analysed alongside 475 CipR.MSM5 isolates from Australia (*n* = 231), France (*n* = 101), Belgium (*n* = 112), and the USA (*n* = 31) (as these two genotypes t10.377 and CipR.MSM5/3.6.1.1.2 were 96% concordant (see Methods). These 943 isolates were constructed into a SNP-based phylogeny with improved resolution and the population was further divided into BAPS clusters (Fig. 2). This analysis supports the suggestion of clonal replacement of the CipR.MSM5 subclade over time, and geographic admixing within the BAPS clusters suggests that the evolution of MSM-associated *S. sonnei* was structured temporally rather than geographically (Fig. 2). Temporal analysis of the outbreak cluser showed

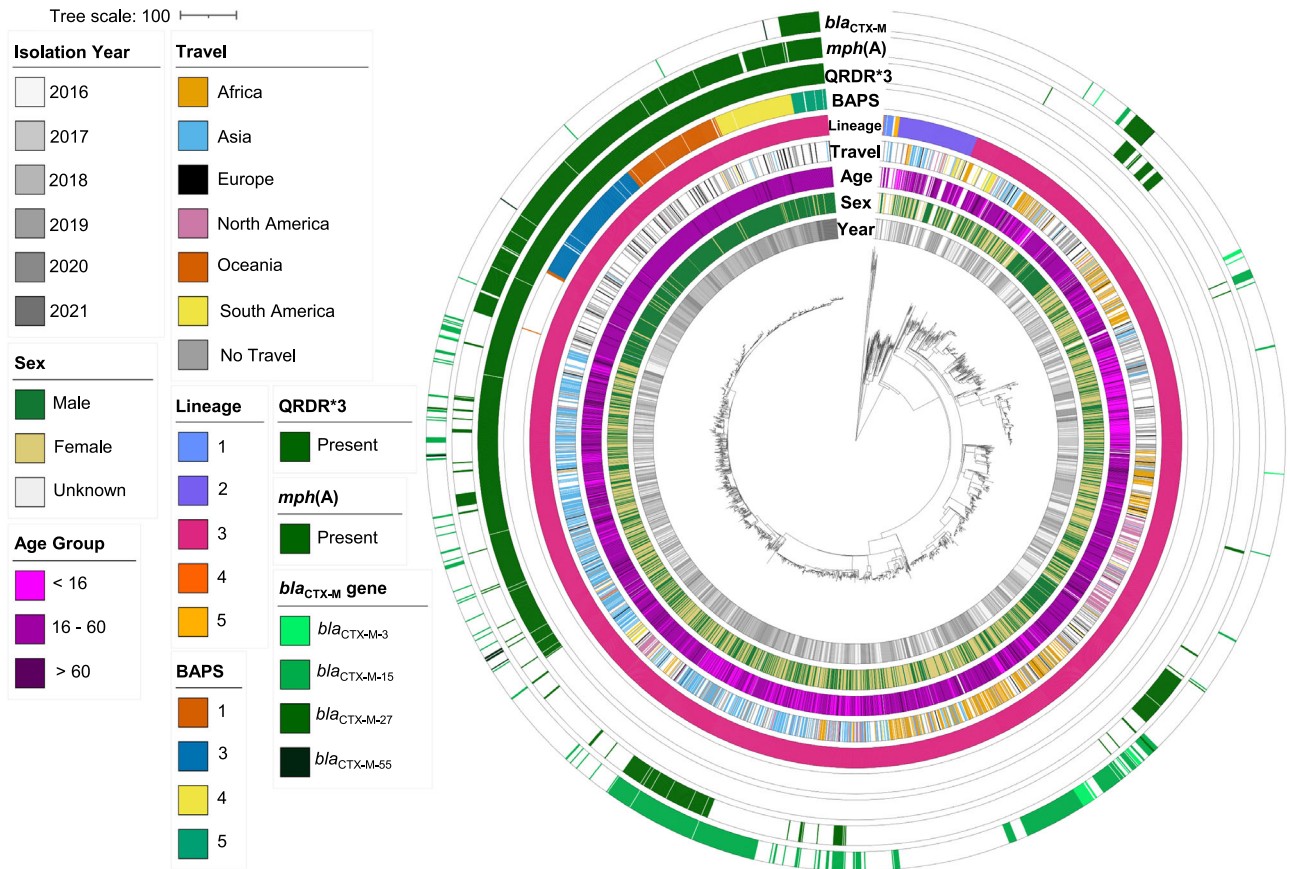

**Tree scale: 100**

**Isolation Year**
- 2016
- 2017
- 2018
- 2019
- 2020
- 2021

**Travel**
- Africa
- Asia
- Europe
- North America
- Oceania
- South America
- No Travel

**Sex**
- Male
- Female
- Unknown

**Age Group**
- < 16
- 16 - 60
- > 60

**Lineage**
- 1
- 2
- 3
- 4
- 5

**BAPS**
- 1
- 3
- 4
- 5

**QRDR*3**
- Present

***mph*(A)**
- Present

***bla*CTX-M gene**
- $bla_{CTX-M-3}$
- $bla_{CTX-M-15}$
- $bla_{CTX-M-27}$
- $bla_{CTX-M-55}$

**Fig. 1 | The emergence of an XDR *S. sonnei* outbreak in the United Kingdom.** A cgMLST dendrogram (midpoint rooted) of clinical isolates from the UK ($n = 2,820$) and genomic subtype references ($n = 120$) with the scale bar indicating distance in cgMLST alleles. Metadata tracks show patient and genomic features for isolates coloured according to the inlaid keys. Specifically, from inner to outer, the patient data comprises: year of isolation, patient sex, age group, and travel history (for UK isolates only), with missing/unavailable/not determined data shown as white. Genomic features then show: isolate lineage, BAPS clusters (for isolates belonging to the t10.377 cluster only), and the presence of mutations in the Quinolone Resistance Determining Region (QRDR*3 denotes all three canonical mutations; *gyrA*_D87G, *gyrA*_S83L and *parC*_S80I) and *mph*(A), and $bla_{CTX-M}$ genes, where white indicates absence of the gene.

that the CipR.MSM5 genotype likely emerged in 2014, and that the BAPS 5 cluster emerged in 2018 (Supplementary Fig. 4).

The clustering of CipR.MSM5 isolates carrying $bla_{CTX-M-27}$ from all countries in BAPS5 indicates that this recently emerged monophyletic lineage of *S. sonnei* was circulating intercontinentally across regions historically considered low-risk for shigellosis. To investigate the evolution of the XDR phenotype in the BAPS5 outbreak cluster, AMR gene profiles were determined for isolates in BAPS1-5 from the UK, Belgium, and NSW, as sets of isolates from other locations included only those with a $bla_{CTX-M}$ gene, (Table 1). This was complemented by phenotypic confirmation for a subset of 14 UK isolates from the CipR.MSM5 subclade (Supplementary Table 1). All isolates were MDR and acquisition of $bla_{CTX-M-27}$ in the BAPS cluster 5 correlated with phenotypic ceftriaxone resistance. BAPS5 isolates lacked $bla_{TEM-1}$ compared with other BAPS, but as expected this did not affect their resistance profile against the antibiotics tested (Supplementary Table 1).

The resistance profiles against the recommended therapeutic antimicrobials (azithromycin, ciprofloxacin, and third-generation cephalosporins) were broadly consistent across BAPS clusters 1–4. Although sharing similar levels of azithromycin resistance with these clusters, the BAPS5 outbreak cluster distinguished itself early in its evolution by acquiring *qnrB19*, and losing $bla_{TEM-1}$ (Fig. 2, Table 1). Notably, *qnrB19* (found on a 2579 bp plasmid) in BAPS5 seems redundant as all isolates in the broader CipR.MSM5 subclade have QRDR mutations that conferred full resistance to ciprofloxacin (MIC ≥ 4 µg/mL), with no increase in MIC associated with the presence of *qnrB19* (Supplementary Table 1). Following the acquisition of this

*qnrB19* plasmid, the BAPS5 cluster subsequently gained $bla_{CTX-M-27}$ which confers resistance to ceftriaxone (MICs ≥12 µg/mL, Supplementary Table 1).

## The $bla_{CTX-M-27}$ containing plasmid, p893816

Analysis of hybrid assemblies from eight outbreak isolates from the UK revealed that all contained a near identical ~83.4 kbp FII plasmid that carries $bla_{CTX-M-27}$. Plasmids from single isolates from France and NSW also match the plasmid from the UK and a BLASTn search revealed that all of these plasmids are also near identical to p893816, reported in MSM-associated CipR.MSM5 *S. sonnei* in London in 2020[28]. This plasmid carries several other AMR genes (Table 1, Supplementary Figs. 2, 3) interspersed with transposon fragments and insertion sequences that may facilitate their movement (Fig. 3a). The plasmid backbone encodes conjugation and stabilisation machinery and carries *psiB* (plasmid SOS inhibition), recently shown to inhibit the disruptive host SOS response to subliminal inhibitory concentrations of ciprofloxacin[19].

Mapping of Illumina reads from BAPS 5 isolates from all countries ($n = 233$) revealed that coverage was 96.1–98.7% for most with $bla_{CTX-M-27}$-containing isolates ($n = 161/184$, 87.5%), including those with assembled plasmids virtually identical to p893816 (Table 2, Supplementary Fig. 5) coverage. Coverage for the remaining BAPS 5 isolates with $bla_{CTX-M-27}$ and for those without this gene ($n = 49$) was distinctly lower (81–93%; Supplementary Fig. 5). This suggests that a p893916-like plasmid carries $bla_{CTX-M-27}$ in the majority of BAPS 5 isolates, but that less related plasmid(s) are present in a minority.

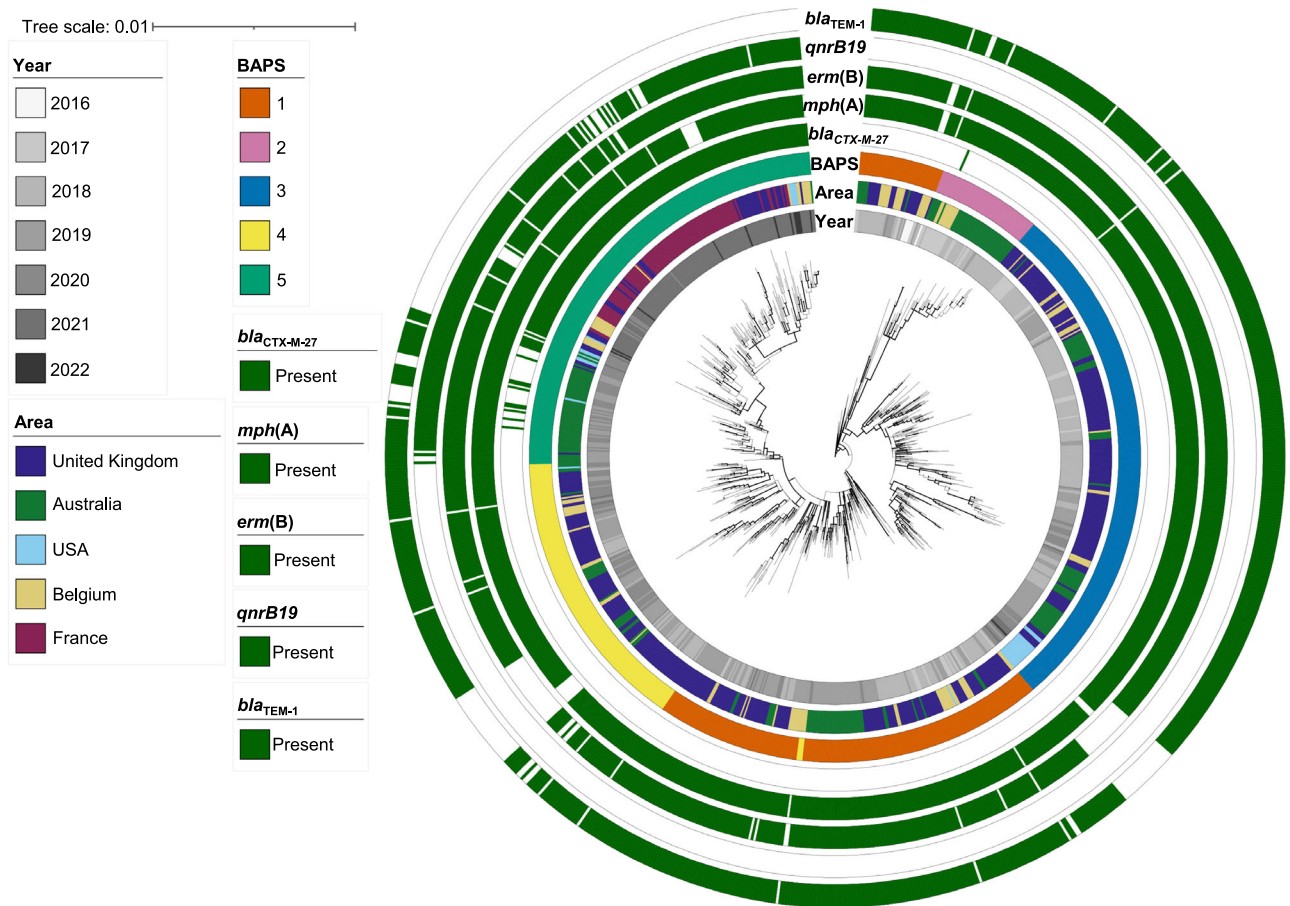

**Fig. 2 | The evolution and international spread of MSM-associated *XDR S. sonnei*.** A midpoint rooted maximum likelihood phylogenetic tree shows the distribution of UK isolates (belonging to both CipR.MSM5 and the t10.377 outbreak cluster, *n* = 468) and relevant related international isolates belonging to CipR.MSM5 (Supplementary Fig. 1, *n* = 475). Metadata tracks show year and country (area) of isolation, BAPS subtype, and the presence of selected AMR genes according to the inlaid keys. The scalebar is provided by IQTree, and represents expected number of substitutions per site across a 1717 bp alignment. Bold branches represent a bootstrap value of ≥ 70 out of 100.

**Table 1 | Proportion of isolates from the UK, Belgium and NSW carrying different AMR genes by BAPS cluster**

| Antimicrobial class | Gene[a] | Proportion of BAPS cluster containing gene (%) | | | | |
|---|---|---|---|---|---|---|
| | | 1 (*n* = 240) | 2 (*n* = 53) | 3 (*n* = 249) | 4 (*n* = 148) | 5 (*n* = 118) |
| Aminoglycosides | *aadA5* | 97.08 | 100 | 97.99 | 76.35 | 100 |
| | *aph(3")-lb/strA* | 87.92 | 98.11 | 85.94 | 79.05 | 83.05 |
| | *aph(6)-ld/strB* | 87.92 | 98.11 | 85.94 | 79.05 | 83.05 |
| β-lactams | *bla*TEM-1 | 97.08 | 94.34 | 97.19 | 76.35 | 36.44 |
| Cephalosporins | *bla*CTX-M-27 | 0 | 1.89 | 0 | 0 | 58.47 |
| Fluroquinolones[b] | *qnrB19* | 0 | 0 | 0 | 0 | 94.92 |
| Macrolides | *erm*(B) | 97.50 | 100 | 99.60 | 93.92 | 100 |
| | *mph*(A) | 97.08 | 100 | 97.99 | 93.92 | 100 |
| Sulphonamides | *sul1* | 89.17 | 98.11 | 85.94 | 79.05 | 84.75 |
| | *sul2* | 87.92 | 96.23 | 85.94 | 79.73 | 83.05 |
| Tetracyclines | *tet*(A) | 100 | 100 | 100 | 100 | 100 |
| Trimethoprim | *dfrA17* | 97.08 | 100 | 97.99 | 81.08 | 100 |

[a]All isolates also carry *sat2* and *dfrA1* and the intrinsic chromosomal *ampC* (annotated as *blaEC*), *acrF* and *emrD* genes.
[b]All isolates have *gyrA*_D87G, *gyrA*_S83L and *parC*_S80I QRDR mutations associated with quinolone resistance.

These plasmids also match Plasmid 3 from an Australian *S. flexneri 3a* isolated even earlier in 2018 (Table 2). Finding the same plasmid in another species suggests it is mobilising among *Shigella*, consistent with previous work showing that p893816 had close relatives in another circulating genotype of MSM-associated *S. sonnei*, MSM2/ VN.KH1[29], and echoing the global expansion of the azithromycin resistance pKSR100-like plasmids among different *Shigella* species[17,19].

To determine the extent and spread of close relatives of p893816 among bacteria, we compared the plasmid sequence with publicly available data. Firstly, comparisons (using BLAST) of the

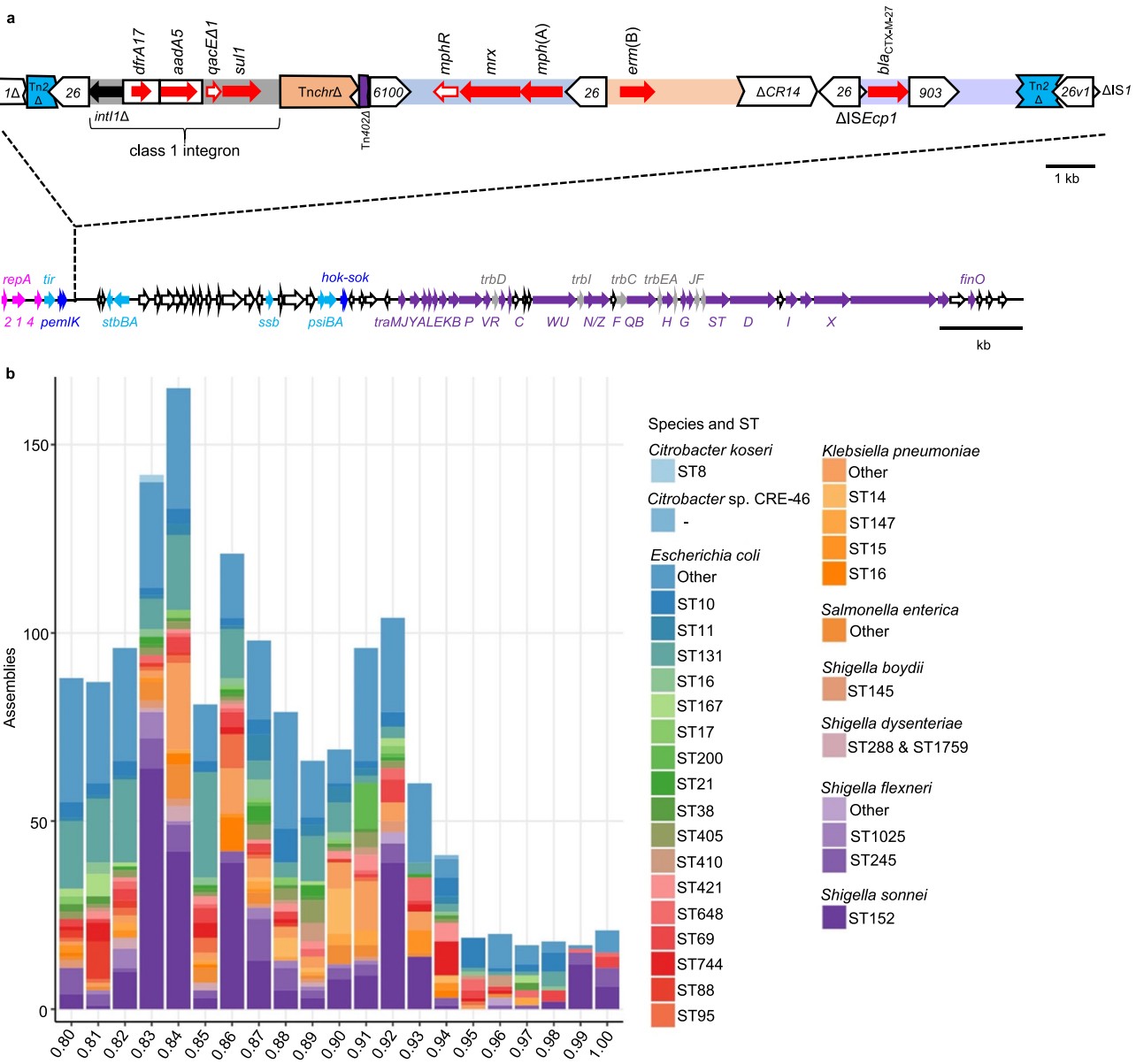

**Fig. 3 | Organisation and distribution of p893816-like plasmids. a** Organisation of p893816-like plasmids. The lengths and directions of transcription of various genes are shown as arrows. The upper track shows the resistance region. AMR genes indicated by solid red arrows, with resistance-related genes outlined in red. The boxes around *dfrA17* and *aadA5* represent the gene cassettes that carry these genes. Block arrows in white indicate various insertion sequences (IS), named/ numbered as labelled, and the direction of their transposase gene. Fragments of different transposons (Tn) are shown as interrupted blocks in different colours (blue, purple, orange) and labelled by name, and a class 1 integron is also shown in grey and labelled. The lower track shows the plasmid backbone, with the AMR region insertion site indicated by dashed lines. Replication genes are in pink, conjugation genes in purple (tra) and grey (trb), toxin-antitoxin systems in blue, other genes encoding known plasmid functions in cyan and genes encoding hypothetical proteins in white. **b** The distribution of p893816-like plasmids. Number of draft genome assemblies in the 661 K COBS data structure showing k-mer similarity (≥0.80) to p893916, coloured by species and sequence type according to the inlaid keys. ST that are represented 10 or more times are named, while rare sequence types are collapsed into the category of 'other' within the respective species.

p893816 sequence with the nucleotide collection at the National Centre for Biotechnology Information (NCBI) revealed multiple similar sequences (≥95% identity, Supplementary Fig. 3). Secondly, a broader search for bacterial isolates containing elements of p893816 was performed using an indexed data structure created from 661,405 uniformly assembled bacterial genomes that were present in the European Nucleotide Archive in November of 2018 (see methods). The index was queried with the constituent k-mers (short sequences of length k) of p893816, identifying datasets that contained ≥80% of the query k-mers; or ≥80% k-mer similarity. It is important to note that

k-mer similarity drops more rapidly than nucleotide identity, as for each SNP difference between the query and the nearest sequence in the index, a window of overlapping k-mers is absent. This search confirmed that some *Shigella* and *E. coli* genomes had sequences of high k-mer similarity and that some more distantly related *Enterobacteriaceae* harboured more distant p893816 relatives (Fig. 3b). A limitation of this analysis is that the index was built from a snapshot in 2018, when this plasmid appears to have emerged, and will miss relationships to more recently deposited sequences. This highlights the value of dynamic ongoing surveillance and how rapid analysis of the

**Table 2 | Details of selected p893816-like plasmids compared in this study (and see Supplementary)**

| Plasmid | p893816 | p1538171_3 | p202008564-6 | p20-001-0088_F | Plasmid-3 |
|---|---|---|---|---|---|
| Accession | MW396858.1 | CP104412 | OP038290 | CP115395 | LR861790 |
| Strain name | 893816 | 1538171 | 202008564 | 20-001-0088 | AUSMDU0022017 |
| Length (bp) | 83,397 | 83,416 | 83,397 | 83,389 | 83,425 |
| Species | *S. sonnei* | *S. sonnei* | *S. sonnei* | *S. sonnei* | *S. flexneri* |
| Subtype | CipR.MSM5 | ND | CipR.MSM5, BAPS 5 | CipR.MSM5, BAPS 5 | Serotype 3a |
| Year | 2020 | 2021 | 2020 | 2019 | 2018 |
| Country | UK | UK | France | Australia | Australia |
| Reference | Locke et al.[28] | This study | This study | This study | Publicly available data |

distribution of mobile genetic elements would further our understanding of transmissible AMR.

As our previous investigations suggested that pKSR100 (GenBank accession CP090162), an epidemiologically successful *Shigella* AMR plasmid, conferred a lower fitness cost on the bacterial host than another less successful co-circulating AMR plasmid, pAPR100 (GenBank accession CP090161)[19], we explored the fitness cost of the p893816-like plasmid p1538171_3 in a naïve *E. coli* host. Analogous experiments here based on area under the curve analysis of growth curves showed that p1538171_3 had a similarly low fitness cost as pKSR100 (Fig. 4). Minimum inhibitory concentration experiments confirmed functional expression of the AMR genes present on the p1538171_3 plasmid (Supplementary Table 2). Unfortunately, this portends that this MDR-plasmid may readily spread throughout MSM-associated *Shigella* globally, perhaps in further MDR strains that have already acquired ciprofloxacin resistance.

## Discussion

Antimicrobial use for treatment of STIs among the at-risk MSM community[30], the possibility of prolonged infection in this group, and the propensity of *Shigella* to acquire genes through horizontal gene transfer[31,32], creates conditions where these pathogens can hone their AMR repertoire. MDR and XDR strains have evolved to become problematic within the MSM community, creating a significant burden with respect to potential treatment failure, and both persistent and reoccurring infections[33]. In 2004, the first UK outbreak of sexually transmitted shigellosis was caused by *S. sonnei*, though relatively few of the isolates carried azithromycin resistance genes such as *mph*(A)[34]. A spate of similar MSM-associated outbreaks of MDR *S. sonnei* occurred across other countries at this time, and perhaps represented an early example of international dissemination of *Shigella* throughout the global MSM community, which has been subsequently observed for MDR subtypes, including azithromycin resistant *Shigella*[18] and now the BAPS5 lineage described in this study. The acquisition of resistance against recommended treatments for shigellosis and other STIs is a global public health concern, and potentially herald an increased risk of treatment failure[35], and eventual emergence of pan-drug resistant *S. sonnei*.

The drivers and trajectory of this emergent XDR lineage, which is additionally ceftriaxone resistant compared with previously described internationally disseminated MSM-associated *Shigella*, are likely to be similar to previously identified phenomena. Specifically, evolution of resistance to third-generation cephalosporins is likely being driven by bystander resistance resulting from shifting treatment regimens for other sexually transmissible illnesses. While historical use of azithromycin for treating many STIs led to the acquisition of *mph*(A) and *erm*(B) among MSM associated *Shigella*[18], the current first-line treatment recommendation for gonorrhoea in the USA, UK, France, and Belgium is now third-generation cephalosporins (e.g. ceftriaxone)[36–39]. Treatment for ongoing high rates of gonorrhoea in the UK[40] thus likely

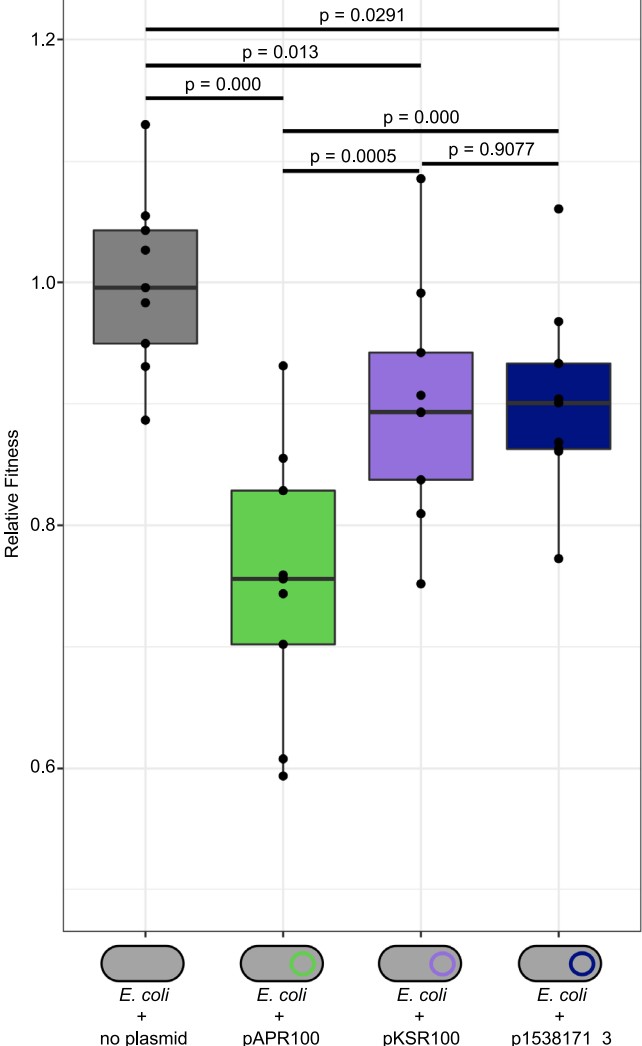

**Fig. 4 | Relative fitness of different *Shigella* MDR plasmids in *E. coli*.** Relative fitness of *E. coli* MG1655 carrying a previously characterised azithromycin resistance plasmid (pAPR100, green; pKSR100, purple), or p1538171_3 (navy) compared with a plasmid-free strain (grey). Each dataset is represented by a box and whisker plot indicating maximum and minimum values (upper and lower limits, respectively), and interquartile ranges (25th to 75th percentiles) of the original data, overlaid with black dots, indicating individual data points. The centre emboldened line on each box represents the median value. The results of pairwise statistical comparisons are shown above and with two-tailed *p*-values (two sample t-tests) indicated above black lines joining the pair comparators. The sample size for statistical purposes is *n* = 3 biologically independent samples (see methods).

contributed to the acquisition of resistance to third-generation cephalosporins into this circulating ciprofloxacin resistant lineage of *S. sonnei*. It is also possible that plasmid fitness cost played a role in the emergence of this XDR strain. Given the global transmission of the low the fitness cost MDR pKSR100-type plasmids among various serotypes of *Shigella*[19], it is likely that the ESBL producing p893816-like plasmids will continue to spread via horizontal gene transfer to other species and serotypes of *Shigella* in MSM, and early evidence of this is emerging in the United Kingdom. (K. Thorley, personal communication 2023).

The ongoing emergence of highly MDR and XDR *Shigella* sublineages conferred by low-fitness plasmids among MSM highlights important future directions for the genomic surveillance of AMR and sexually transmissible enteric infections (STEIs) in this patient group, and beyond. Firstly, given the evidence for antimicrobial selection pressures driving the evolution of AMR in STEIs, the impact of STI treatments on AMR could be monitored more holistically through active surveillance of the microbiota in this risk group (e.g. through detection of AMR genes in stool and/or other sentinel organisms). Active surveillance like this could feed into evidence-based updates of treatment guidelines for STIs to also monitor and mitigate the further development of XDR, and PDR strains of STEIs. Secondly, collaborative genomic epidemiology studies such as ours highlight that the emergence of a concerning resistance phenotype in MSM associated *Shigella* in one country may lead to rapid global dissemination within a short time frame (i.e., a decade) and efforts to develop, coordinate, and harmonise, global genomic surveillance in line with the new WHO strategy in this area[41] are imperative to counter the public health threat posed by AMR STEIs. The final lesson from our study stems from this XDR outbreak lineage evolving from a clade that dominated MSM-associated shigellosis for nearly a decade, and a plasmid detected as early as 2018, highlighting a window of opportunity to have pre-empted this new AMR threat. To ensure global genomic surveillance efforts can eventually transition to a prospective, predictive framework where preventative action can be taken, and intervention efforts focused, we must work toward identifying early indicators for AMR threats with a high potential for global spread.

## Methods

### Reference isolates

Representatives of all *S. sonnei* subclades, including 120 reference isolates representative of a global genotyping framework (accessions in Supplementary Data 1)[14] were included in the core genome Multi-Locus Sequence Typing (cgMLST) tree in this study to contextualise the UK outbreak.

### Isolates and data from the United Kingdom Health Security Agency (UKHSA)

All *S. sonnei* isolates with an assigned Single Nucleotide Polymorphism (SNP) address in England (97%, $n = 2895$ of 2982 submitted) received by the United Kingdom national reference laboratory (Gastrointestinal Bacteria Reference Unit) between 01/01/2016 and 30/12/2021 were included in this study. The SNP address is a linkage-clustering system in which reference-mapping based pairwise distances at 250, 100, 50, 25, 10, 5, and 0 SNP are numbered and listed in this order, joined by periods[42].

Data on patient age was available for all isolates, data on patient sex for 99% ($n = 2875/2895$), and explicitly declared information about travel for approximately half ($n = 1487/2895$, 51%). In total, 429 isolates were from patients who reported recent travel to Asia, 289 to Africa, 73 to Europe, 121 to North America, 6 to Oceania, 38 to South America, and 73 to an unknown location. There were 458 isolates from patients where 'no travel' was explicitly indicated and for 1408 isolates information from patients was missing. In the absence of individual patient risk factors on sexual activity being available, possible MSM associated

isolates were classified by proxy as deriving from men aged 16 years or older, without a history of recent international travel, in line with a previous validation of this approach in this setting[43].

All isolates were routinely sequenced (Illumina) according to previously described protocols[44]. Isolates were assigned SNP addresses, according to in house subtyping methods[42]. The SNP address of relevance here is 1.1.1.1.377. This lineage became XDR and the outbreak was investigated at the level of the 10-SNP threshold, so the 1.1.1.1.377 SNP linkage cluster is referred to here as t10.377 for consistency with the initial report[26]. Among the 2895 *S. sonnei* isolates in this study, 483 belonged to SNP linkage cluster t10.377.

To enable exploring international links to the UK outbreak, we employed a recently described global genotyping framework[14]. This indicated that the UK-based SNP linkage cluster t10.377 roughly approximated the international 3.6.1.1.2 genotype of *S. sonnei* (96% concordance statistic, Supplementary Data 1). The human-readable nomenclature for this genotype is CipR.MSM5, due to its known ciprofloxacin resistance and previous description in MSM as 'Clade 5', the prevailing *S. sonnei* subtype in the UK[6,14].

Eight plasmid sequences from UK isolates (see accessions in Supplementary Data 1) generated from hybrid assemblies with Oxford Nanopore technology generated as previously[45], were used in plasmid comparisons.

### International outbreak associated isolates

CipR.MSM5 *S. sonnei* genotype isolates, with or without $bla_{\text{CTX-M-27}}$, from public health surveillance organisations in other countries were integrated into the analysis (based on data volume and sharing arrangements). A brief overview for each country and aggregate data on patient age and gender, which were consistent with isolates deriving from MSM-associated transmission, is provided here.

**Belgium.** All *S. sonnei* confirmed as CipR.MSM5 isolated at Sciensano's reference laboratory between January 2017 and March 2022 and with sequence data were included. All ($n = 112$) isolates came from patients aged up to 84 years who were mostly male ($n = 101/112$, 90%). Sequence data was obtained using the following protocol: isolates were cultured overnight in BHI broth (BD) at 37 °C. DNA was extracted using an MgC Bacterial DNA Kit™ with a 60 µL elution volume (Atrida, Amersfoort, The Netherlands), following the manufacturer's instructions. Sequencing libraries were constructed using the Illumina Nextera XT DNA sample preparation kit and sequenced on an Illumina MiSeq instrument with a 250 bp paired-end protocol (MiSeq v3 chemistry), according to the manufacturer's instructions (Illumina, San Diego, CA, USA).

**France.** All CipR.MSM5 isolates that contained $bla_{\text{CTX-M-27}}$ ($n = 101$) isolated at the Enteric Bacterial Pathogens Unit at the Institut Pasteur between 01/09/2020 and 15/02/2022 were included in this study. The isolates came from patients from 13 to 68 years and the majority were male ($n = 98/101$, 97%). These were sequenced using the following protocol: Genomic DNA was extracted with the MagNA Pure DNA isolation kit (Roche Molecular Systems, Indianapolis, IN, USA), in accordance with the manufacturer's instructions. Whole-genome sequencing was performed with the Mutualized Platform for Microbiology (P2M) at Institut Pasteur, Paris. The libraries were prepared with the Nextera XT kit (Illumina, San Diego, CA, USA) and sequencing was performed with the NextSeq 500 system.

The sequence of a single plasmid (p202008564-6) from isolate 202008564 was obtained. Genomic DNA was prepared as follows: the isolates were cultured overnight at 37 °C in alkaline nutrient agar (20 g casein meat peptone E2 from Organotechnie; 5 g sodium chloride from Sigma; 15 g Bacto agar from Difco; distilled water to 1 L; adjusted to pH 8.4; autoclaved at 121 °C for 15 min). A few isolated colonies from the overnight culture were used to inoculate 20 mL of Brain-Heart-Infusion

(BHI) broth and were cultured until a final $OD_{600}$ of 0.8 was reached at 37 °C with shaking (200 rpm—Thermo Scientific MaxQ 6800). Bacterial cells were harvested by centrifugation and the DNA extraction was performed by two different methods. For isolate 202008118, we followed the protocol described by von Mentzer et al.[46], except that MaXtract High Density columns (Qiagen) were used (instead of phase lock tubes) and DNA was resuspended in molecular biology grade water (instead of 10 mM Tris pH 8.0). We used a Genomic-tip 100/G column (Qiagen) according to the manufacturer's protocol. The library was prepared according to the instructions of the Native barcoding genomic DNA (with EXP-NBD104, EXP-NBD114, and SQK-LSK109) procedure provided by Oxford Nanopore Technology. Sequencing was then performed on a MinION Mk1C apparatus (Oxford Nanopore Technologies). The genomic sequences of the isolates were assembled from long and short reads, with a hybrid approach and Unicycler (v 0.4.8)[47]. A polishing step was performed with Pilon (v 1.23), to generate a high-quality plasmid sequence[48], p202008564-6 (GenBank accession no. OP038290).

**United States of America.** Data for all publicly available CipR.MSM5 isolates which contained a gene in the $bla_{CTX-M}$ family ($n = 31$) and associated age and gender data were kindly provided by PulseNet, Centre for Diseases Control. Aggregate metadata showed that these isolates came from patients aged between 20 and 79 years that were predominately male ($n = 30/31$, 97%).

**Australia.** From the state of Victoria, sequences of CipR.MSM5 $bla_{CTX-M-27}$ isolates ($n = 3$) from routine health surveillance (Microbiological Diagnostic Unit, Public Health Laboratory, Doherty Institute) were analysed, two of which have been described previously (AUSMDU00040532 and AUSMDU00040564)[23]. All three were from male patients aged between 16 and 60 years. From the state of New South Wales, all CipR.MSM5 isolates collected during routine surveillance October 2017-November 2020 ($n = 232$) were included. Isolates were from patients between 19 and 83 years (mean = 40 years) and mainly male ($n = 224/232$, 97%). Genomic DNA was extracted from pure cultures using a QIAGEN DNeasy Blood and Tissue Mini Kit (QIAGEN). Sequencing libraries were prepared using Nextera XT DNA Library Prep Kit (Illumina) and sequenced on a NextSeq500 at the Microbial Genomics Reference Laboratory, Westmead Hospital. Isolate 20-001-0088 was used for long-read sequencing. DNA was extracted from growth on Horse Blood agar (Thermofisher) using DNeasy® Ultra-Clean® Microbial Kit (Qiagen) with mechanical lysis reduced to 2 min. DNA quality and quantity were assessed on a Nano-300 Micro-Spectrophotometer (Allsheng) and a Qubit™ 2 Fluorometer (Life Technologies) while integrity was assessed by 0.6% (w/v) agarose gel electrophoresis. A library was prepared using the SQK-RBK004 Rapid barcoding kit (Oxford Nanopore Technologies) according to manufacturer's instructions, loaded into a R4.9.1 flow cell and sequenced on a MinION Mk 1B (Oxford Nanopore Technologies). Sequencing was performed for up to 24 h and base-calling was performed post sequencing using Guppy (v 3.4.5 + f1fbfb)[49]. Hybrid assembly was performed using Unicycler (v 0.4.8)[47]. The assembly was further short-read polished on Pilon (v 1.23)[48], using BAM files generated by Mini-Map2 (v 2.17-r941)[50], until no further changes could be made. The assembly was automatically annotated using Prokka (v 1.14.6)[51], running on metagenome mode and using genetic code 11.

**Bioinformatic analyses**

**Clustering dendrogram of UK isolates.** A dendrogram was constructed for the UK isolates available on Enterobase ($n = 2820/2895$) and the global context representatives ($n = 120$) using the NINJA NJ tree algorithm (v 1.0) on the cgMLST hierarchical clustering results from the algorithm implemented in Enterobase (v 1.1.3)[52,53]. The tree was visualised using the Interactive Tree of Life (iTOL, v 5)[54].

**Phylogenetic reconstruction of the international outbreak.** To explore international relationships, we constructed a tree from international CipR.MSM5 isolates ($n = 475$) alongside isolates belonging to the UK outbreak cluster ($n = 468$ of 482 which generated pseudogenomes of uniform length, t10.377)[26]. Illumina paired end reads for the isolates from the UK were downloaded from the Sequence Read Archive (SRA) using fastq-dump (SRA toolkit v 2.11.0)[55] with default settings except the '--split 3' parameter was used to automatically separate paired end read data into two separate FASTQ files. Nucleotide sequences were trimmed using Trimmomatic (v 0.39)[56] with the parameters: Leading: 20, Trailing: 20, SlidingWindow:4:20, MinLen:40, and quality checked with MultiQC (v 1.12)[57]. Burrows-Wheeler Aligner (BWA v 0.7.17) was used with default parameters to align the *S. sonnei* sequences to *Shigella sonnei* 53G as the reference genome (NCBI accession number: *GCA_00283715.1_ASM28371v1*)[58]. PICARD (v 2.27.2)[59] was used to mark and remove artificial nucleotide duplicates from the genomes of the *S. sonnei* isolates using default parameters. SAMTOOLS (v 1.11) was used to index the files[60]. BCFTOOLS (v 1.9)[61], was used to call and filter variants, using default parameters and '*c --ploidy 1 −O -o*' for variant calling and '*--SnpGap 15 --IndelGap 15*' for filtering[62]. A chromosomal pseudogenome was created for all isolates and prophages and plasmids were masked using sed (v 4.2.2)[63]. Gubbins (v 3.2.1)[64] was then used to remove remaining regions of recombination using default parameters except: '*c 30, f 60*'. Multiple sequence alignments were then used to 1) impute a phylogenetic tree using IQTREE (v 2.2.0.3)[65] with the default parameters except: '*-ntmax 25 -bb 1000 -m GTR + I + R + ASC*, and 2) determine population clusters using Rhier Bayesian Analysis of Genetic Population Structure (RhierBAPS v 1.01)[66,67].

**Temporal signal analyses.** Newick tree files from iTOL (v 5)[54], derived from the IQTREE (v 2.2.0.3)[65] output described above for isolates where the full dates were known (day, month and year, UK, $n = 555$, NSW $n = 232$), were imported into TempEst[68], (only the year of isolate collection is listed in Supplementary Data 1, for patient confidentiality reasons). Default parameters were used, with the function set to 'heuristic residual mean squared', and with 'best-fitting root' activated. Dates were parsed as a calendar date (DD/MM/YYYY). Dates were specified as 'years' and 'since some time in the past' and lineage 'root-to-tip' data used to infer the date of emergence of the most recent common ancestors.

**Genome assembly and identifying genes of interest.** Draft genomes were assembled for isolates from all locations using Unicycler (v 0.5.0)[47], using the 'forward paired', 'reverse paired', 'forward unpaired', 'reverse unpaired' trimmed and quality controlled FASTQ files. Draft genome assemblies were checked for quality using QUAST (v 5.02)[69]. Default parameters were used except '*min_fasta_length 200,--vcf*' were used instead. Draft genomes were then analysed for AMR genes using NCBI AMRFinder Plus (v 3.10.24)[70] with the organism set as *Escherichia*, and the '--plus' parameter to obtain information on stress genes. SonneiTyping Script (v 20210201)[71] embedded in Mykrobe (v 0.11.0)[72] was used for genotyping and retrieval of QRDR mutations in *gyrA* and *parC* to infer ciprofloxacin resistance[73].

**Statistical testing.** Standard proportions were calculated, and statistical support for phylogenetic analyses evaluated using chi-square testing. Plasmid fitness experiments were evaluated by two sample *t* test.

**In silico plasmid analyses.** The eight UK IncFII plasmids were re-orientated to start with the *repA* gene using circulator Fixstart (v 1.5.5)[74], followed by annotation using Prokka (v 1.14.6),[51]. Plasmids (GBK files) were compared (>95% nucleotide identity) and visualised using Clinker (v 0.0.21)[75]. For a broader search, the sequence of p893916 was queried against a Compact Bit-sliced Signature (COBS)

index[76], of 661,405 curated draft genomes[77], with a k-mer similarity cut-off of 80%. Genomes that were high quality and had a species ID of 0.8 or higher (*n* = 1505) were included in the distribution figure. Species and sequence types (ST) were extracted from Supplementary file File4_QC_characterisation_611K[77]. But due to the difficulty in distinguishing *E. coli* and *Shigella* spp., some species initially identified as *E. coli* were updated to the relevant *Shigella* species based on the assigned ST. A BLAST Ring Image Generator (BRIG, v 0.95)[78], plot was constructed using default settings with p893816 (GenBank accession MW396858) as the reference plasmid genome. Illumina reads for all BAPS 5 isolates were were mapped against p893816 using Burrow-Wheeler Aligner (v 0.7.17-r118)[58]. Percentage coverage at 1X sequencing depth calculated by Qualimap (v 2.2.2)[79] with default parameters. Isolates known to have plasmids virtually identical to p893816 (Table 2, Fig. S2) gave values of 96-97%.

**In vitro plasmid analyses.** The p893816-like plasmid from *S. sonnei* clinical isolate 1538717 (p1538717_3, CP104412) was conjugated into *E. coli* MG1655 and used in antimicrobial susceptibility testing and relative fitness assays, all as previously described[19]. *E. coli* MG1655 carrying pKSR100 or pAPR100 were generated from isolates obtained from the UKHSA, as previously described[19]. *E. coli* MG1655 and derivatives that each carry a different plasmid were inoculated into M9 minimal media supplemented with thiamine (10 μg/ml) and grown overnight with shaking at 220 rpm at 37 °C. These pre-cultures were adjusted to $OD_{600}$ 0.05 in a total volume of 1 ml and 150 μl aliquots inoculated into individual wells of a 96-well plate (Greiner Bio One, UK). The 96-well plate was then incubated with shaking at 37 °C in a Synergy H1 multi-mode plate reader (BioTek Instruments) taking optical density measurements at 600 nm every 15 min. Data were analysed using the R package Growthcurver and Area Under the Curve (AUC) output was used to calculate the relative fitness of each strain carrying the plasmid compared with the plasmid-free strain[80]. Each experiment consisted of three technical replicates and was repeated three times (i.e. three biological replicates). Details of all strains and plasmids used in this study can be found in Supplementary Table 3.

**Antimicrobial susceptibility testing.** To confirm resistances inferred from sequence data, 14 UK isolates with representative genotypic AMR profiles, and *E. coli* MG1655 with and without p1538717_3 underwent antimicrobial susceptibility testing using Liofilchem® gradient MIC Test Strips (MTS) according to manufacturer's instructions to determine the minimum inhibitory concentrations (MICs). The following antimicrobial and MIC ranges (all in μg/mL) were used: ciprofloxacin (CIP, 0.002–32), ertapenem (ERT, 0.002–32), mecillinam (MEC, 0.016–256), azithromycin (AZM, 0.016–256), ceftriaxone (CRO, 0.002–32), trimethoprim sulfamethoxazole (SXT, 0.002–32), gentamicin (GEN, 0.016–256). Results were interpreted as defined by the European Committee on Antimicrobial Susceptibility Testing (EUCAST)[81].

**Bioinformatics packages and services used.** A summary of all bioinformatics packages and services used in this study, alongside information about commands and versions used, can be found in Supplementary Table 4 and the Reporting Summary document.

**Ethics statement**
No individual patient consent was required or sought as UKHSA has authority to handle patient data for public health monitoring and infection control under section 251 of the UK National Health Service Act of 2006 (previously section 60 of the Health and Social Care Act of 2001).

**Reporting summary**
Further information on research design is available in the Nature Portfolio Reporting Summary linked to this article.

## Data availability

Sequence data relating to all isolates has been deposited in the Sequence Read Archive under BioProject numbers PRJNA315192 (United Kingdom), PRJEB44801 (France), PRJEB40097 (Belgium), and PRJNA613115 (Australia, New South Wales). Individual isolate accession numbers and isolate metadata are available in the Supplementary Data 1 as indicated in the text. Correspondence regarding isolates from the United Kingdom and the overall study should be directed to Dr. Claire Jenkins and Professor Kate Baker. Correspondence regarding isolates: from France should be directed to Professor François-Xavier Weill; from Belgium should be directed to Pieter-Jan Ceyssens; from the United States of America should be directed to Kaitlin Tagg or PulseNet; and from Australia to Ben Howden and Danielle Ingle (Victoria) or Elena Martinez or Jonathan Iredell (New South Wales).

## Code availability

No custom code was written to conduct this study. Details of bioinformatic packages and commands used can be found in Supplementary Table 4.

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

## Acknowledgements

This study was funded by the National Institute for Health and Care Research (NIHR) Health Protection Research Unit in Gastrointestinal Infections, a partnership between the UK Health Security Agency, the University of Liverpool, and the University of Warwick. The views expressed are those of the author(s) and not necessarily those of the NIHR, the UK Health Security Agency or the Department of Health and Social Care. The authors thank Hattie Webb, Kaitlin Tagg, Jason Folster, Jean Whichard, and other staff at the United States Centres for Disease Control for helpful discussions to guide access to genome data from PulseNet. The authors are also grateful to pathology laboratories that provided isolates for genomic surveillance. This work was also supported by MRC and BBSRC grants held by KSB (MR/R020787/1 and BB/V009184/1, respectively).

## Author contributions

L.C.E.M., C.J., and K.B. were responsible for the writing of the original draft. All authors contributed to the review and editing of the manuscript. L.C.E.M., D.G., L.C., S.P., E.M., G.B., C.C., M.D.S., J.D., E.S., E.N., C.J., and K.S.B. were responsible for formal analysis. L.C.E.M., D.G., L.C., S.P., E.M., G.B., C.C., M.D.S., J.D., A.G., I.S., E.S., V.S., J.I., S.L., C.J., and K.S.B. were responsible for the investigation. L.C.E.M., D.G., S.P., E.M., G.B., J.D., A.G., E.S., C.J., K.S.B. were responsible for the methodology. L.C.E.M., F.X.W., P.J.C., C.J., K.S.B. were responsible for conceptualisation. D.G., E.M., J.D., A.G., D.I., S.L., E.N., and C.J. were responsible for data curation. D.G., I.S., V.S., J.I., B.P.H., S.L., E.N., F.X.W., C.J., and K.S.B. were responsible for providing resources. M.D.S. was responsible for validation. V.S., J.I., B.P.H., F.X.W., C.J., K.S.B. were responsible for funding acquisition. V.S., J.I., C.J. and K.S.B. were responsible for project administration. R.J.B., V.S., J.I., C.J., and K.S.B. were responsible for supervision. L.C.E.M., D.G., S.P., G.B., C.C., and M.D.S. were responsible for visualisation.

## Competing interests

The authors declare no competing interests.
