## [Peer Review File · Nature Communications]

The evolution and international spread of extensively drug resistant *Shigella sonnei*REVIEWER COMMENTS

Reviewer #1 (Remarks to the Author):

Overall

This is a well-designed experiment that significantly broadens, in original ways, the findings in a recent publication involving two of the authors.

Charles, et al. 2022. Outbreak of sexually transmitted, extensively drug-resistant *Shigella sonnei* in the UK, 2021–22: a descriptive epidemiological study. *The Lancet Infectious Diseases*.

This manuscript is generally well written, and the content is original and noteworthy. It demonstrates that there is significant ongoing evolution in the pKSR100-like plasmid lineage that now harbors an ESBL, leading to significant complications for treatment. They also demonstrate that this is a particular concern among MSM. The cited material adds significantly to interpretation.

I have recommended one additional experiment to show the transfer of AMR phenotypes along with the plasmid. The strain for this experiment already exists, and should confirm that the plasmid genes contribute to the phenotypes attributed to it in the paper.

The conclusion needs some more significant revision, but mostly by subtraction.

I recommend acceptance of the paper after revision and would like to thank the authors for their excellent work on this emerging and high-priority pathogen. Seeing an ESBL in the MSM-associated pKSR100 plasmid group, known for its MDR phenotype and antibiotic resistance cassettes, is alarming and needs more attention.

Introduction

The introduction is generally well written and sufficiently covers the topics brought up in the paper. There are a few specific issues, outlined below, that are easily addressed. While there is no standard definition for XDR *Shigella*, one ought to be included for clarity.

Line 60-61

Citation 10 does not support sentence as written.

The year 2012 is not from citation 6, but a paper that it cites, found here. The proportions shown are ages 16-60, excluding child diagnoses.

<https://www.eurosurveillance.org/content/10.2807/1560-7917.ES2015.20.15.21097>

Field et al. 2015. Intensified shigellosis epidemic associated with sexual transmission in men who have sex with men - *Shigella flexneri* and *S. sonnei* in England, 2004 to end of February 2015. *Eurosurveillance*.

Lines 65-68

Citation 15 should be moved up to the sentence beginning "For Example" (line 66).

Line 66: This sentence is "For example", but it is unclear how it is an example of the utility of splitting strains into different sublineages (line 65).

As written, it is unclear how the four-lineage classification system for *S. sonnei* has contributed to AMR research, and only lineage 3 is mentioned in this paragraph.

Lines 81-83

A clear definition for XDR *Shigella* should be given here. The term is not clearly defined in citation 10, and some clarity is needed as to what specifically is being claimed regarding available treatments. Is "ciprofloxacin, azithromycin and extended-spectrum β -lactam antibiotics (e.g., ceftriaxone)" the definition, if so, why these antibiotics specifically (ie resistance to all first-line antibiotics, resistance to all but 2 first- and second-line antibiotics, etc.).

The recent article (Citation 25) from related data defines it similarly to the 2012 proposal from ECDC and CDC.

Charles, et al. 2022. Outbreak of sexually transmitted, extensively drug-resistant *Shigella sonnei* in the UK, 2021–22: a descriptive epidemiological study. *The Lancet Infectious Diseases*.

Magiorakos, et al. 2012. Multidrug-resistant, extensively drug-resistant and pandrug-resistant bacteria: an international expert proposal for interim standard definitions for acquired resistance. *Clinical Microbiology and Infection*.

Materials and Methods

This section is well constructed and should allow for the replication of the experiments with a few minor revisions.

Lines 98-99

Which 120 reference isolates? Please provide an accession for each in a supplementary file, or cite the supplementary material file if those are included therein.

Define cgMLST.

Lines 115-118

SNP address is used earlier (line 102) but defined and cited here. Are these the same method, or is this order as intended? The earlier section, reference isolates, could simply be moved below to correct.

Line 130

I would like to applaud the care and effort put into this section by the authors.

Line 266

Is there a reference for the conjugal transfer assay, or are the conditions new to this study? Include a citation or methods.

Results and Discussion

This section is well constructed and, for the most part, only requires minor edits. Respectfully, since the plasmid has been successfully transferred, I would request an antibiogram be performed on MG1655 p893816. While they have shown the transferal of genes, they should be able to confer the transferal of phenotype, or perhaps reveal the requirement of a *Shigella* specific factor for full resistance. Also suitable would be a normally susceptible *S. sonnei* or *S. flexneri* strain.

Apart from this, the section only needs minor revisions, minor additions, and a few points of clarification and/or explanation. It is original, high-quality and noteworthy.

Line 301

Suggestion: "reflecting the different bacterial populations endemic to these regions."

Line 302-305

The authors should consider combining this language with the previous sentence. For example: "... regions, a well-known phenomenon.[citations]".

There needs to be a p-value at line 304 if the authors are claiming that the proportion of international travel to Asia, Africa, and Latin America is lower than the rest of the strains in their UK clinical isolate collection.

Figure 2

Consider changing the color scheme for the outer bands. Bright green is hard to read. Non-essential for acceptance.

Line 336

Specify the number of isolates from each country here

Line 359-360

"... seems somewhat superfluous" should be replaced for more concrete language. Suggestion:

"seems redundant"

Line 362-364

This section about qnrB19 is a bit too speculative, even for discussion materials. Its apparent lack of phenotypic contribution has been stated, the authors should at least shorten to a simple statement that the gene and/or plasmid may have an unknown function, perhaps beyond antimicrobial resistance, if not remove the sentence entirely. Otherwise, more evidence is needed and available. For instance, is there evidence that qnrB19 or plasmid is or is becoming vestigial, such as the accumulation of mutations?

Line 385-386

Authors should mention other fluoroquinolone resistance mechanisms found in *Shigella*, such as mutations in gyrA/B.

Table 2

p2020-001-0088_F needs an accession before final acceptance.

Line 390-391

This sentence does not read well. I figured out what it means, but a less informed reader will not. Please clarify.

Line 390-399

I am confused on what "kmer similarity" represents. Is it sequence identity? Linear sequence coverage? Exact kmer matches? Something else? Additionally, COBS is not defined as an acronym, nor is it prevalent enough to warrant inclusion without its expanded form.

The citation on the Github page appears to be an arxiv with a different title than citation 58.

At the very least, a short explanation is needed in the results of what COBS is measuring since it is not a prevalent technique.

See: <https://arxiv.org/abs/1905.09624>

Bingmann, et al. 2019. COBS: a Compact Bit-Sliced Signature Index. arXiv 1905.09624

Line 394

Remove parenthesis. I may be mistaken on this note after a clarification of COBS.

Line 396

Replace interrogation with: experiment, method, analysis, etc.

Figure 3A

A legend is required here, or the description for the upper section needs to match the drawings in the figure. The red arrows are in the diagram, not above. There is a white box around dfrA17 and aadA5 of unknown function. There are two blue Tn2Δ that are not mentioned. There are several unexplained colored regions in the area. Was TnchrΔ supposed to be a white box?

Figure 3B

Reinforcing a comment from earlier, defining "Kmer [kmer?] similarity" in the body of the text will facilitate interpretability of this figure. My current understanding is that it shows pKSR100-like plasmids are relatively prevalent among a variety of related bacteria. kmer similarity here representing the relative sequence similarity among kmers of the original sequence.

Figure 4

Suggestion: Group the 4 bars into 3 separate significance categories: a, b, c, and mark each bar with the corresponding letter. This also allows a significant difference between MG1655 alone and MG1655 pKSR100/p893816 to be shown, if that exists.

Currently, the figure does not show a significant difference between MG1655 p893816 and MG1655 pAPR100, though I assume one exists based on the other results here, and needs to be

stated as it is the focus of the manuscript.

Conclusions

This section has some redundant elements, particularly the last two paragraphs both highlighting the need for international surveillance. There is substantial introductory material here, which should either be moved or cut. Line 450-451 was a particular concern.

First paragraph (Lines 416-428)

Suggestion: much of the material here is covered in the introduction, or is introductory in nature, and without specific references to the findings in the paper. The second paragraph (Line 430) is a much stronger setting of stakes that relies on the scientific findings.

Line 418

Choose setting, environment, or something else (conditions, etc.).

Lines 420-422

This is a highly fragmented sentence with too many commas, and a comma hanging off the end.

Lines 444-447

"with a third-generation cephalosporin now recommended, with ceftriaxone being the preferred first choice." This fragment is a little unclear as to which countries are now suggesting this. Are all now recommending ceftriaxone as the preferred first-line drug?

More broadly, is treatment for gonorrhea the only disease for which therapy recommendations have changed, or has it been part of a broader movement away from azithromycin? Or, are the authors suggesting that just the gonorrhea treatment recommendation alone is prevalent enough to drive evolution in *Shigella*? I would find that plausible and worth mention if gonorrhea case rates are high, which requires a citation of that fact, if it has not been cited already.

Lines 450-451, 451-453

450-451: It is unclear what is meant by "holistic" management, including on what level the authors are advocating "active surveillance of the commensal gut microbiota." Should individuals be tested for antibiotic resistance genes in their stool? Wastewater monitoring? As a part of STI testing?

451-453: In contrast to the above, this sentence is perfectly supported and clear. I recommend removing the preceding sentence unless it is substantially clarified.

Line 466-468

"As we have seen with the SARS-CoV-2 pandemic, near-global harmonized genomic surveillance is possible" needs at least a reference, and may not be the best example for this manuscript. I tend to agree with the sentiment, but I can think of several scientists who would point to lower sequencing prevalence in key areas from where variants emerged. Something more specific about how to expand the worthwhile efforts of the research in this manuscript, and their potential impact, might be warranted instead, perhaps with an eye towards other STIs, or HAIs.

Reviewer #2 (Remarks to the Author):

The manuscript titled "The evolution and international spread of extensively drug resistant *Shigella sonnei*" describes the origins and dissemination of a sub-clade of *S. sonnei* that acquired blaCTX-M-27. The authors also detail the genetic context of the antibiotic resistance determinants and fitness costs of the plasmid that contains the blaCTX-M-27 gene. This study is well done and provides important insight into the emergence and spread of an extensively drug resistant *S. sonnei*. Also, this study demonstrates the importance of global collaboration to track the spread of antibiotic resistant pathogens that pose a threat to public health.

Line 102: Please describe SNP address here as this appears to be the first mention of it. The sentence at Lines 117-118 does a nice job of this and could be moved up.

Line 336/Figure 2: Please clarify the international isolates included in Figure 2. Are the international isolates all part of the t10.377 linkage cluster?

Lines 368-371: Is a p893816-like plasmid present among all of the t10.377/CipR.MSM5 that contain blaCTX-M-27?

Table 2: Is an accession number available for p2020-001-0088_F?

Lines 394-395: Do these E. coli and other Shigella genomes with significant plasmid similarity harbor the same resistance genes as those on p893816, particularly blaCTX-M-27?

Line 430: There is awkward wording in this sentence. Please clarify.

Reviewer #3 (Remarks to the Author):

In this manuscript, the authors present an analysis of several thousand *Shigella sonnei* genomes and metadata. From my reading, the outbreak is associated with a specific sub-clade of *S. sonnei*, possessing an MDR FII plasmid, which is broadly distributed amongst bacterial species based on the 661K COBS database, and has a limited cost to fitness in a heterologous bacterial system.

My feeling is that this study reports an important observation - namely, the rapid emergence of an XDR phenotype in *S. sonnei*, well-correlated to the acquisition of a resistance plasmid, in a sub-lineage of *S. sonnei* Lineage 3. It is important that these sorts of findings be highlighted, to provide genetic explanations of clinically-important phenotypes. I do feel that some additional analysis - such as a dated phylogeny - would lend strength to the authors' findings and lend further support to their claims, and I have elaborated on this below.

General comments

The title is extremely general, and to my reading, this manuscript describes a single sub-lineage of *S. sonnei* which has evolved to become XDR through the acquisition of a resistance plasmid. I suggest that the title be modified to describe more specifically the phenomena being reported in this manuscript; as written, it seems too general - or that it describes all possible ways in which XDR phenotypes arise in *S. sonnei* rather than the more targeted observations that the text reports. HOWEVER, if it's the case that this is the first and only XDR report in *S. sonnei*, then this is an important thing to state - but might require re-wording and re-prioritising of aspects of the text.

It was not clear to me, particularly from the introduction, exactly why the authors embarked on this (very substantial) genomic analysis. Perhaps the last paragraph in the Introduction could be expanded to set out more explicitly the link between the observed outbreaks of *S. sonnei* in multiple countries, and the present study? What is the benefit of using genomics to study this problem?

Did the authors manage to use the metadata and phylogeny for dating the emergence of the XDR lineage and/or the plasmid acquisition event? If not, please consider including a BEAST analysis or similar, given the year of isolation for many isolates is known, and there does seem to be structure in the topology of the tree in Figure 2.

It wasn't clear to me if the sequencing reported in the manuscript was done as part of this study, or if it was routine genomic surveillance which was leveraged to draw these conclusions. If the latter, this might explain the relative paucity of details in lab and sequencing methods for the UK isolates in particular (reliant on citations instead). Please could the authors elaborate on this?

Specific comments

Line 32 - "natural history" - I have to admit that I did not understand what it was the authors were referring to with this statement. Please re-word.

Methods throughout - please double check if two hyphens have been autocorrected to a long dash.

Please add a table of strains, plasmids, genotypes etc for the materials used in the laboratory experiments; fitness experiments etc.

Fitness experiment - from the description in Methods, I do not understand how relative fitness has been estimated for these strains and I have a lot of questions about this approach. Are all strains grown, the AUC computed, and compared back to those of the wild-type isolate? If so, what is the purpose of using a GFP-expressing MG1655 derivative, particularly since only OD600 measurements are taken by the Synergy plate reader rather than fluorescence? Are the plasmids maintained under selection? If not, how is their loss within the population of the cells in the microtitre plate quantified/assessed? Why have the authors chosen MG1655 rather than working with a *Shigella* lab strain? Since the authors place a lot of emphasis on the similarities between this plasmid and pKSR100 in terms of it having a relatively low fitness cost compared to other conjugative plasmids, I would greatly appreciate more details being included here.

Figure 1 - I might be misinterpreting this figure, but lineages don't seem to be assigned to the vast majority of isolates? I understand that several of the other metadata entries are specific to certain sub-groups of isolates (as in the legend), but wouldn't lineage assignment be possible to do for the whole dataset? The reason I ask the question is that the authors make statements about the majority of isolates belonging to Lineage 3 (line 287), but to my eye, this is not obvious from this figure. Please consider revising this.

Figure 2 - Please add a scale bar to this figure and number of isolates. Please italicise and subscript text in the legend where appropriate. Please could the specifics of this phylogeny also be clarified (or the reader be pointed to a relevant part of the text) - specifically, how was the alignment from which this tree was built computed? 1,717 sites seems like a very small number for a *Shigella* phylogeny, and while I accept that this is an outbreak clone, the topology of this tree might be misleading or might benefit from the inclusion of bootstrap data. For instance there seems to be a BAPS4 isolate nestled within a large clade of BAPS1 isolates.

Line 339 - "harmonise isolate inclusion across sites". I don't understand what the authors mean here, please could this be re-worded?

Line 362 - "poor genetic streamlining". Again, I don't understand what the authors mean here, please could this be re-worded?

Figure 3 - the sequence types in Fig 3B legend - were these metadata assigned by the authors, or are they derived from the COBS database?

Figure 4 - please could datapoints be added to this plot, since based on my reading of the Methods, there aren't all that many replicates to visualise?

Line 418 - "setting/environment" - perhaps best to choose one or the other? I suggest 'environment' but leave the decision to the authors.

Paragraph beginning line 441 - the authors indulge in a lot of speculation in this paragraph, referring to several "likely" scenarios. This is not necessarily a bad thing in the Conclusions, but as written, it does make it hard to discern between statements which are reasonable deductions given the data, and those which may be more conjecture. Perhaps some re-phrasing of this paragraph would strengthen the authors' opinions and conclusions?

Line 455 - "rapid global dissemination". I agree with the authors, and I think this is my main concern with the text as written. Having read the manuscript several times, I didn't come away

with a strong feeling of the importance of this point which, to my mind, is a really important observation and conclusion. My opinion is that this should be emphasised more unambiguously, including in the abstract (perhaps including dates/times etc), to highlight the strengths of the dataset and observations which the authors present.

Line 462 - 'international cooperation'. Agreed. I would recommend adding some citations to relevant literature here - e.g., commentary articles around data sharing from COVID-19, PulseNet LAC, PHA4GE, other networks etc.

Figure S1 - please add a note to the legend explaining the curved lines on the right-hand-side, for the benefit of a reader. I assume these are repeats/transposase genes, but certainly could be wrong. Please clarify.

Author's Response: Bold

General:

We would like to thank the reviewers for their time and helpful comments on the manuscript, which has led to significant improvements. NB: 'New L' refer to Line numbers in the revised manuscript, 'Old L' refers to the original manuscript submission.

REVIEWER COMMENTS

Reviewer #1 (Remarks to the Author):

Overall

This is a well-designed experiment that significantly broadens, in original ways, the findings in a recent publication involving two of the authors.

Charles, et al. 2022. Outbreak of sexually transmitted, extensively drug-resistant *Shigella sonnei* in the UK, 2021–22: a descriptive epidemiological study. *The Lancet Infectious Diseases*.

This manuscript is generally well written, and the content is original and noteworthy. It demonstrates that there is significant ongoing evolution in the pKSR100-like plasmid lineage that now harbors an ESBL, leading to significant complications for treatment. They also demonstrate that this is a particular concern among MSM. The cited material adds significantly to interpretation.

I have recommended one additional experiment to show the transfer of AMR phenotypes along with the plasmid. The strain for this experiment already exists and should confirm that the plasmid genes contribute to the phenotypes attributed to it in the paper.

We note that this has been completed (further details below).

The conclusion needs some more significant revision, but mostly by subtraction.

This has also been done (see below).

I recommend acceptance of the paper after revision and would like to thank the authors for their excellent work on this emerging and high-priority pathogen. Seeing an ESBL in the MSM-associated pKSR100 plasmid group, known for its MDR phenotype and antibiotic resistance cassettes, is alarming and needs more attention.

We thank the reviewer for this positive feedback on our work.

Introduction

The introduction is generally well written and sufficiently covers the topics brought up in the paper. There are a few specific issues, outlined below, that are easily addressed. While there is no standard definition for XDR *Shigella*, one ought to be included for clarity.

This has also been done (see below re: Old L81 – 83).

Line 60-61 ;

Citation 10 does not support sentence as written.

The year 2012 is not from citation 6, but a paper that it cites, found here. The proportions shown are ages 16-60, excluding child diagnoses.

<https://www.eurosurveillance.org/content/10.2807/1560-7917.ES2015.20.15.21097>

Field et al. 2015. Intensified shigellosis epidemic associated with sexual transmission in men who have sex with men - *Shigella flexneri* and *S. sonnei* in England, 2004 to end of February 2015. *Eurosurveillance*.

We agree with the reviewer and have modified the sentence and inserted the appropriate citation (now reference 10).

Lines 65-68

Citation 15 should be moved up to the sentence beginning "For Example" (line 66).

Thank you, this has now been fixed (New L 69).

Line 66: This sentence is “For example”, but it is unclear how it is an example of the utility of splitting strains into different sublineages (line 65). As written, it is unclear how the four-lineage classification system for *S. sonnei* has contributed to AMR research, and only lineage 3 is mentioned in this paragraph.

Thank you, the paragraph has been edited for clarity to highlight how specific clades and their relationship with AMR have been tracked using whole genome sequencing (New Ls 67 – 71).

Lines 81-83

A clear definition for XDR *Shigella* should be given here. The term is not clearly defined in citation 10, and some clarity is needed as to what specifically is being claimed regarding available treatments. Is “ciprofloxacin, azithromycin and extended-spectrum β -lactam antibiotics (e.g., ceftriaxone)” the definition, if so, why these antibiotics specifically (ie resistance to all first-line antibiotics, resistance to all but 2 first- and /second-line antibiotics, etc.).

The recent article (Citation 25) from related data defines it similarly to the 2012 proposal from ECDC and CDC.

Charles, et al. 2022. Outbreak of sexually transmitted, extensively drug-resistant *Shigella sonnei* in the UK, 2021–22: a descriptive epidemiological study. *The Lancet Infectious Diseases*.

Magiorakos, et al. 2012. Multidrug-resistant, extensively drug-resistant and pandrug-resistant bacteria: an international expert proposal for interim standard definitions for acquired resistance. *Clinical Microbiology and Infection*.

Thank you, we have now added a clear definition of XDR *S. sonnei*. We have defined XDR *S. sonnei* as being resistant to the only first-line antimicrobial (ciprofloxacin) and to all-but-one of the second-line antimicrobials (pivmecillinam, ceftriaxone, azithromycin) recommended by the World Health Organisation (WHO) for treatment of shigellosis. (See New Ls 82 – 85).

Materials and Methods

This section is well constructed and should allow for the replication of the experiments with a few minor revisions.

Lines 98-99

Which 120 reference isolates? Please provide an accession for each in a supplementary file or cite the supplementary material file if those are included therein.

We thank the reviewer for highlighting this omitted citation of the supplementary materials. The 120 reference isolate accessions are available in Supplementary Table 1, which has now been made clear in the text (New L 104).

Define cgMLST.

This has now been defined in the text (New Ls 104 – 105).

Lines 115-118

SNP address is used earlier (line 102) but defined and cited here. Are these the same method, or is this order as intended? The earlier section, reference isolates, could simply be moved below to correct.

We thank the reviewer for highlighting this readability issue. ‘SNP address’ has now been defined and cited at its first point of being mentioned (see New Ls 108, 110 - 112).

Line 130

I would like to applaud the care and effort put into this section by the authors.

Thank you for your kind words.

Line 266

Is there a reference for the conjugal transfer assay, or are the conditions new to this study? Include a citation or methods.

We thank the review for highlighting this omission - we have now included a citation to the relevant background paper from our group where we have used the conjugal transfer assay from which two *E. coli* strains carrying pKSR100 and pAPR100 have originated (reference 19, New L 272).

Results and Discussion

This section is well constructed and, for the most part, only requires minor edits. Respectfully, since the plasmid has been successfully transferred, I would request an antibiogram be performed on MG1655 p893816. While they have shown the transferal of genes, they should be able to confer the transferal of phenotype, or perhaps reveal the requirement of a *Shigella* specific factor for full resistance. Also suitable would be a normally susceptible *S. sonnei* or *S. flexneri* strain.

We thank the reviewer for raising this important aspect of phenotypic functionality of the AMR genes in the new host of p893816. We have now performed minimum inhibitory concentration (MIC) determinations on *E. coli* MG1655 strain with and without p893816 to confirm that the phenotype arises from the acquisition of AMR genes carried on p893816 and the results are displayed in the new Supplementary Table 3.

Apart from this, the section only needs minor revisions, minor additions, and a few points of clarification and/or explanation. It is original, high-quality and noteworthy.

Line 301

Suggestion: "reflecting the different bacterial populations endemic to these regions."

Line 302-305

The authors should consider combining this language with the previous sentence. For example: "... regions, a well-known phenomenon.[citations]".

We thank the reviewer for these suggestions, which we have taken (New Ls 317 - 318).

There needs to be a p-value at line 304 if the authors are claiming that the proportion of international travel to Asia, Africa, and Latin America is lower than the rest of the strains in their UK clinical isolate collection.

This has now been calculated (p-value <0.0001) and shown next to raw values in the text (New Ls 318 - 320).

Figure 2

Consider changing the color scheme for the outer bands. Bright green is hard to read. Non-essential for acceptance.

Thank you, this has been changed to a darker shade of green (Figure 2).

Line 336

Specify the number of isolates from each country here

This has been added to the text (New Ls 343 – 346).

Line 359-360

"... seems somewhat superfluous" should be replaced for more concrete language. Suggestion: "seems redundant"

Thank you, this has been changed to the suggested wording (New L 367).

Line 362-364

This section about qnrB19 is a bit too speculative, even for discussion materials. Its apparent lack of phenotypic contribution has been stated, the authors should at least shorten to a simple statement that the gene and/or plasmid may have an unknown function, perhaps beyond antimicrobial resistance, if not remove the sentence entirely. Otherwise, more evidence is needed and available. For instance, is there evidence that qnrB19 or plasmid is or is becoming vestigial, such as the accumulation of mutations?

Thank you, we have reduced this down to a single sentence addressing the presence of *qnrB19* and its apparent redundancy (see edits at New L 366 - 369).

Line 385-386

Authors should mention other fluoroquinolone resistance mechanisms found in *Shigella*, such as mutations in *gyrA/B*.

We have elected to change the wording to refer to the phenotype ('CipR.MSM5') rather than underlying genetic mechanisms (see New L 377).

Table 2

p2020-001-0088_F needs an accession before final acceptance.

This has been added to Table 2.

Line 390-391

This sentence does not read well. I figured out what it means, but a less informed reader will not. Please clarify.

This paragraph has been heavily edited for clarity to address the reviewer's concerns (New Ls 397 – 400).

Line 390-399

I am confused on what "kmer similarity" represents. Is it sequence identity? Linear sequence coverage? Exact kmer matches? Something else? Additionally, COBS is not defined as an acronym, nor is it prevalent enough to warrant inclusion without its expanded form.

The citation on the Github page appears to be an arxiv with a different title than citation 58.

At the very least, a short explanation is needed in the results of what COBS is measuring since it is not a prevalent technique.

See: <https://arxiv.org/abs/1905.09624>

Bingmann, et al. 2019. COBS: a Compact Bit-Sliced Signature Index. arXiv 1905.09624

We thank the reviewer for these suggestions for clarity. The citation has now been corrected in the methods (New L 257, reference 61) and a greater elaboration of what the implications of kmer similarity are added to the text in the results (New Ls 403 – 408).

Line 394

Remove parenthesis. I may be mistaken on this note after a clarification of COBS.

Thank you, the parenthesis has been removed.

Line 396

Replace interrogation with: experiment, method, analysis, etc.

Thank you, 'interrogation / interrogated' has been changed to 'analysis / analysed' for all instances of its use throughout the manuscript (see edits throughout).

Figure 3A

A legend is required here, or the description for the upper section needs to match the drawings in the figure. The red arrows are in the diagram, not above. There is a white box around dfrA17 and aadA5 of unknown function. There are two blue Tn2Δ that are not mentioned. There are several unexplained colored regions in the area. Was TnchrΔ supposed to be a white box?

We appreciate the reviewer's request for greater clarity here and have added further detail to the Figure legend which now better defines the background to all colours and symbols in the Figure (see Figure 3A updated legend).

Figure 3B

Reinforcing a comment from earlier, defining "Kmer [kmer?] similarity" in the body of the text will facilitate interpretability of this figure. My current understanding is that it shows pKSR100-like plasmids are relatively prevalent among a variety of related bacteria. kmer similarity here representing the relative sequence similarity among kmers of the original sequence?

We agree with the reviewer that their helpful suggestion to better define kmer similarity in the text has improved readability of this portion of the figure.

Figure 4

Suggestion: Group the 4 bars into 3 separate significance categories: a, b, c, and mark each bar with the corresponding letter. This also allows a significant difference between MG1655 alone and MG1655 pKSR100/p893816 to be shown, if that exists.

Currently, the figure does not show a significant difference between MG1655 p893816 and MG1655 pAPR100, though I assume one exists based on the other results here, and needs to be stated as it is the focus of the manuscript.

We thank the reviewer for this suggestion to improve the information displayed in Figure 4. We have now carried out separate t-tests for each of the datasets and modified the figure to include additional information about the statistical significance between all relevant pairs, including those highlighted above (see new Figure 4 and edited legend at New Ls 438 - 443).

Conclusions

This section has some redundant elements, particularly the last two paragraphs both highlighting the need for international surveillance. There is substantial introductory material here, which should either be moved or cut. Line 450-451 was a particular concern.

We thank the reviewer for this specific and constructive summary and have heavily edited this section in line with the comments from this, and other reviewers (see New Ls 436 – 441).

First paragraph (Lines 416-428)

Suggestion: much of the material here is covered in the introduction, or is introductory in nature, and without specific references to the findings in the paper. The second paragraph (Line 430) is a much stronger setting of stakes that relies on the scientific findings.

We have edited this to make clearer that this paragraph exists to appropriately contextualise this study (New Ls 442 – 455).

Line 418

Choose setting, environment, or something else (conditions, etc.).

Thank you, this has been changed to conditions (New L 445).

Lines 420-422

This is a highly fragmented sentence with too many commas, and a comma hanging off the end.

Thank you, this sentence has now been re-written to remove the excessive commas (New Ls 447 - 449).

Lines 444-447

“with a third-generation cephalosporin now recommended, with ceftriaxone being the preferred first choice.” This fragment is a little unclear as to which countries are now suggesting this. Are all now recommending ceftriaxone as the preferred first-line drug?

This paragraph has been heavily edited for clarity and citations moved to the end of the sentence as all nations mentioned are recommending a third-generation cephalosporin as first-line treatment. (New Ls 462-464).

More broadly, is treatment for gonorrhoea the only disease for which therapy recommendations have changed, or has it been part of a broader movement away from azithromycin? Or, are the authors suggesting that just the gonorrhoea treatment recommendation alone is prevalent enough to drive evolution in *Shigella*? I would find that plausible and worth mention if gonorrhoea case rates are high, which requires a citation of that fact, if it has not been cited already.

Thank you, gonorrhoea case rates remain high (51,074 reported diagnoses in the UK in 2021), which we have now added a sentence about and cited in the text (New L 464).

Lines 450-451, 451-453

450-451: It is unclear what is meant by “holistic” management, including on what level the authors are advocating “active surveillance of the commensal gut microbiota.” Should individuals be tested for antibiotic resistance genes in their stool? Wastewater monitoring? As a part of STI testing?

We have heavily edited the text to make our meaning clearer – see “Given the evidence for antimicrobial selection pressures driving the evolution of STEI resistance, the impact of STI treatments on AMR could be monitored more holistically through active surveillance of the microbiota in this risk group (e.g. through detection of AMR genes in stool and/or other sentinel organisms). Active surveillance like this could feed into evidence-based updates of treatment guidelines for STIs to also monitor and mitigate the further development of XDR, and PDR strains of STEIs.” (At New Ls 474 – 479).

451-453: In contrast to the above, this sentence is perfectly supported and clear. I recommend removing the preceding sentence unless it is substantially clarified.

We have substantially clarified the preceding sentence (see above).

Line 466-468

“As we have seen with the SARS-CoV-2 pandemic, near-global harmonized genomic surveillance is possible” needs at least a reference, and may not be the best example for this manuscript. I tend to agree with the sentiment, but I can think of several scientists who would point to lower sequencing prevalence in key areas from where variants emerged. Something more specific about how to expand the worthwhile efforts of the research in this manuscript, and their potential impact, might be warranted instead, perhaps with an eye towards other STIs, or HAIs.

We agree with the reviewer and in line with their comments and substantial edits made to this section, this suggestion has now been removed.

Reviewer #2 (Remarks to the Author):

The manuscript titled “The evolution and international spread of extensively drug resistant *Shigella sonnei*” describes the origins and dissemination of a sub-clade of *S. sonnei* that acquired blaCTX-M-27. The authors also detail the genetic context of the antibiotic resistance determinants and fitness costs of the plasmid that contains the blaCTX-M-27 gene. This study is well done and provides important insight into the emergence and spread of an extensively drug resistant *S. sonnei*. Also, this study demonstrates the importance of global collaboration to track the spread of antibiotic resistant pathogens that pose a threat to public health.

We thank the reviewer for their positive comments on the manuscript.

Line 102: Please describe SNP address here as this appears to be the first mention of it. The sentence at Lines 117-118 does a nice job of this and could be moved up.

Thank you, the definition of SNP address has now been moved up to the first time it is mentioned in line with this and Reviewer 1’s comments (New Ls 108 and 110 – 112).

Line 336/Figure 2: Please clarify the international isolates included in Figure 2. Are the international isolates all part of the t10.377 linkage cluster?

SNP-access (giving rise to the t10.377 designation) is UK specific and not available for international isolates. The relationship between t10.377 and 3.6.1.1.2 has been clarified extensively throughout the text, including in the Figure 2 legend, and supplemented by a new supplementary Figure (Supplementary Figure 1).

Lines 368-371: Is a p893816-like plasmid present among all of the t10.377/CipR.MSM5 that contain blaCTX-M-27?

We thank the reviewer for this important question and have an additional analysis to confirm that this is the case, which is shown in Supplementary Figure 5 and described in the text at New Ls 383 - 388. Mapping of Illumina reads suggests that a plasmid very closely related to p893616 is present in the majority of BAPS 5 with bla_{CTX-M-27}.

Table 2: Is an accession number available for p2020-001-0088_F?

This has now been added to Table 2.

Lines 394-395: Do these *E. coli* and other *Shigella* genomes with significant plasmid similarity harbor the same resistance genes as those on p893816, particularly blaCTX-M-27?

This is unknown and, while we appreciate the reviewer's point, would represent a substantial additional analysis of public data that is not substantive to the conclusions of the study in light of the results of the already presented in the manuscript and the additional analysis undertaken above.

Line 430: There is awkward wording in this sentence. Please clarify.

The conclusions section has been extensively edited in line with reviewer 1's comments and this is no longer present in the text.

Reviewer #3 (Remarks to the Author):

In this manuscript, the authors present an analysis of several thousand *Shigella sonnei* genomes and metadata. From my reading, the outbreak is associated with a specific sub-clade of *S. sonnei*, possessing an MDR FII plasmid, which is broadly distributed amongst bacterial species based on the 661K COBS database, and has a limited cost to fitness in a heterologous bacterial system.

My feeling is that this study reports an important observation - namely, the rapid emergence of an XDR phenotype in *S. sonnei*, well-correlated to the acquisition of a resistance plasmid, in a sub-lineage of *S. sonnei* Lineage 3. It is important that these sorts of findings be highlighted, to provide genetic explanations of clinically-important phenotypes. I do feel that some additional analysis - such as a dated phylogeny - would lend strength to the authors' findings and lend further support to their claims, and I have elaborated on this below.

We thank the reviewer for this helpful summary and suggestions, which we have taken.

General comments

The title is extremely general, and to my reading, this manuscript describes a single sub-lineage of *S. sonnei* which has evolved to become XDR through the acquisition of a resistance plasmid. I suggest that the title be modified to describe more specifically the phenomena being reported in this manuscript; as written, it seems too general - or that it describes all possible ways in which XDR phenotypes arise in *S. sonnei* rather than the more targeted observations that the text reports. HOWEVER, if it's the case that this is the first and only XDR report in *S. sonnei*, then this is an important thing to state - but might require re-wording and re-prioritising of aspects of the text.

We agree with the reviewer that the title is general, but at the time of reporting, this is the first sustained XDR *S. sonnei* lineage that has been sustained in the UK and demonstrated to transmit internationally. We have updated Figure 1 to better reflect this by incorporating the distribution of CTX-M genes through the phylogeny (see New Figure 1 and text at New Ls 304 – 309). We are also happy to take editorial advice on this and if the editor would prefer, we could potentially update the title to include “among men who have sex with men” at the end.

It was not clear to me, particularly from the introduction, exactly why the authors embarked on this (very substantial) genomic analysis. Perhaps the last paragraph in the Introduction could be expanded to set out more explicitly the link between the observed outbreaks of *S. sonnei* in multiple countries, and the present study? What is the benefit of using genomics to study this problem?

Thank you, a line has been added at the end of the introduction to clarify the drivers behind the study (New Ls 94 – 98).

Did the authors manage to use the metadata and phylogeny for dating the emergence of the XDR lineage and/or the plasmid acquisition event? If not, please consider including a BEAST analysis or similar, given the year of isolation for many isolates is known, and there does seem to be structure in the topology of the tree in Figure 2.

We thank the reviewer for this helpful suggestion. We have now undertaken a temporal signal analysis using TempEst, which predicts the emergence of CipR.MSM5 (3.6.1.1.2) and BAPS 5 as 2014 and 2018, respectively. This has been added to the text at New Ls 232 – 239 (methods), 350 - 351 (results), and shown in Supplementary Figure 4.

It wasn't clear to me if the sequencing reported in the manuscript was done as part of this study, or if it was routine genomic surveillance which was leveraged to draw these conclusions. If the latter, this might explain the relative paucity of details in lab and sequencing methods for the UK isolates in particular (reliant on citations instead). Please could the authors elaborate on this?

The author is correct that the manuscript largely relied on amalgamating routinely generated sequence data (New L 122). Only the nanopore sequencing was specifically conducted for this study.

Specific comments

Line 32 - "natural history" - I have to admit that I did not understand what it was the authors were referring to with this statement. Please re-word.

Thank you, 'natural history' has been changed to 'evolutionary history' (New L 33).

Methods throughout - please double check if two hyphens have been autocorrected to a long dash.

Thank you for noticing this error. All long dashes have been corrected to two hyphens.

Please add a table of strains, plasmids, genotypes etc for the materials used in the laboratory experiments; fitness experiments etc.

Of course – these are now provided in Supplementary Table 4.

Fitness experiment - from the description in Methods, I do not understand how relative fitness has been estimated for these strains and I have a lot of questions about this approach. Are all strains grown, the AUC computed, and compared back to those of the wild-type isolate? If so, what is the purpose of using a GFP-expressing MG1655 derivative, particularly since only OD600 measurements are taken by the Synergy plate reader rather than fluorescence? Are the plasmids maintained under selection? If not, how is their loss within the population of the cells in the microtitre plate quantified/assessed? Why have the authors chosen MG1655 rather than working with a *Shigella* lab strain? Since the authors place a lot of emphasis on the similarities between this plasmid and pKSR100 in terms of it having a relatively low fitness cost compared to other conjugative plasmids, I would greatly appreciate more details being included here.

We thank the reviewer for this query and understand that the chromosomally-encoded GFP gene has caused confusion here. The *E. coli* strain was a gift from another researcher, and it has a chromosomally-encoded GFP as well as kanamycin resistance (now mentioned in Supplementary Table 4). These features were used to differentiate transconjugants from the donor *S. sonnei* strain and was not used in the fitness experiment. Also, any fitness cost arising from those genes did not affect our fitness assay as they were present in all strains used. To make it clearer, we have now removed the description of the GFP and kanamycin resistance genes from the methods section and placed in the strains and plasmids table (Supplementary Table 4) to hopefully make the message clear and avoid any confusion.

The plasmids were not maintained under any selection for the fitness assay as these plasmids have post-segregational killing mechanisms in the form of toxin-antitoxin systems encoded on them which have previously prohibited plasmid loss during long-term subculture experiments.

We have also added further details of the previous and new experiments to the text at New Ls 268 - 281 (methods), 414 - 421 (results), and the new Supplementary Table 4.

While we appreciate the reviewer's concern about testing the plasmid phenotypes in another *Shigella* lab strain, we have so-far been unable to transfer p1538171_3 (the p898316-like plasmid) into our routinely used clinical strain of *Shigella*. We are not certain about the reason behind this,

and the experiment is ongoing, but we do not feel this detracts from the overall message of the paper. We now have another paper in press showing the natural emergence of the same plasmid in *S. flexneri* 2a, so we know the plasmid is moving among these species horizontally in nature. In summary, we feel that measuring the fitness effects of these plasmids in a neutral yet relevant host (i.e., another Enterobacterial species that co-inhabits the gut with *Shigella*) combined with the *S. flexneri* study is equally impactful, especially with no other plasmid systems present to interfere with the plasmids under investigation in the neutral *E. coli* host.

Figure 1 - I might be misinterpreting this figure, but lineages don't seem to be assigned to the vast majority of isolates? I understand that several of the other metadata entries are specific to certain sub-groups of isolates (as in the legend), but wouldn't lineage assignment be possible to do for the whole dataset? The reason I ask the question is that the authors make statements about the majority of isolates belonging to Lineage 3 (line 287), but to my eye, this is not obvious from this figure. Please consider revising this.

We now indicate the lineage for all isolates in Figure 1.

Figure 2 - Please add a scale bar to this figure and number of isolates. Please italicise and subscript text in the legend where appropriate. Please could the specifics of this phylogeny also be clarified (or the reader be pointed to a relevant part of the text) - specifically, how was the alignment from which this tree was built computed? 1,717 sites seems like a very small number for a *Shigella* phylogeny, and while I accept that this is an outbreak clone, the topology of this tree might be misleading or might benefit from the inclusion of bootstrap data. For instance there seems to be a BAPS4 isolate nestled within a large clade of BAPS1 isolates.

We thank the reviewer for the helpful additions which improve clarity. The scale bar produced by IQTree has been added to this tree. The number of isolates has also been added to the figure legend. Italicisation and subscript text have been added as appropriate. Bootstrap values have been added to the tree, as requested, and represented by emboldened thicker lines for branches with a bootstrap value $\geq 70\%$. The reviewer is correct that the two BAPS4 isolates that appear aligned with BAPS1 are on an ancestral branch with poor bootstrap support. 1,717 for the number of sites is indeed a small number, but these isolates are all very closely related outbreak strains and most of the genetic changes seem to be within the accessory genome, which was masked for the purposes of core-genome phylogenetic analyses. This is also analogous to previous similarly disseminated sublineages of MSM associated *Shigella* (e.g., 1160 SNPs in Baker *et al* 2015).

Line 339 - "harmonise isolate inclusion across sites". I don't understand what the authors mean here, please could this be re-worded?

We thank the reviewer for highlighting this confusing wording, we have now moved this to the methods (New Ls 128 – 132) and made clear that our selection of the 3.6.1.1.2 genotype was to enable exploring international links to the UK outbreak, as SNP address is a UK-specific notation.

Line 362 - "poor genetic streamlining". Again, I don't understand what the authors mean here, please could this be re-worded?

This wording has since been removed from the text owing to comments from this, and Reviewer 1.

Figure 3 - the sequence types in Fig 3B legend - were these metadata assigned by the authors, or are they derived from the COBS database?

The sequence types in Fig 3B form part of the metadata of the COBS661K data structure with some additional curation. This has now been made clear in the methods (see New Ls 256 – 262) and in the Fig 3B legend.

Figure 4 - please could datapoints be added to this plot, since based on my reading of the Methods, there aren't all that many replicates to visualise?

Thank you for the constructive suggestion. Individual datapoints have now been added to this plot (see New Figure 4).

Line 418 - “setting/environment” - perhaps best to choose one or the other? I suggest ‘environment’ but leave the decision to the authors.

This was also highlighted by reviewer 1 and we have updated the wording to ‘conditions’ (see New L 445).

Paragraph beginning line 441 - the authors indulge in a lot of speculation in this paragraph, referring to several “likely” scenarios. This is not necessarily a bad thing in the Conclusions, but as written, it does make it hard to discern between statements which are reasonable deductions given the data, and those which may be more conjecture. Perhaps some re-phrasing of this paragraph would strengthen the authors’ opinions and conclusions?

We thank the reviewer who, alongside reviewer 1, suggested improving clarity and readability through the conclusions section. This has now been heavily edited in line with these comments which addresses the reviewers’ concerns (New Ls 442 – 489).

Re-phrase this paragraph to make clear

Line 455 - “rapid global dissemination”. I agree with the authors, and I think this is my main concern with the text as written. Having read the manuscript several times, I didn’t come away with a strong feeling of the importance of this point which, to my mind, is a really important observation and conclusion. My opinion is that this should be emphasised more unambiguously, including in the abstract (perhaps including dates/times etc), to highlight the strengths of the dataset and observations which the authors present.

We appreciate the reviewer’s comment and suggestion to do the temporal analysis. We have now added the most recent common ancestor time (2018) to the abstract (see New L 36).

Line 462 - ‘international cooperation’. Agreed. I would recommend adding some citations to relevant literature here - e.g., commentary articles around data sharing from COVID-19, PulseNet LAC, PHA4GE, other networks etc.

We thank the reviewer for their suggestion. In line with comments from this, and reviewer 1, we have removed the reference to COVID-19 and cite the WHO global genomic surveillance strategy as relevant literature (New Ls 481 – 483, Reference 81).

Figure S1 - please add a note to the legend explaining the curved lines on the right-hand-side, for the benefit of a reader. I assume these are repeats/transposase genes, but certainly could be wrong. Please clarify.

Thank you, an explanation has now been added to the figure legend (Supplementary Figure 2 in the new manuscript) regarding these curved lines, which do indeed represent multiple copies of the transposase of IS26 (*tnpA₂₆*).

REVIEWERS' COMMENTS

Reviewer #1 (Remarks to the Author):

I have reviewed the updated manuscript and have only minor comments, none pertaining to content. My comments from the first draft have been adequately addressed, and this updated manuscript is suitable for publication. Thanks to the authors for this important work.

Minor comments:

Line 65: Citations inside of period.

Line 65/66: A comma after "and further clades and subclades" before citation 14 helps with clarity to offset the descriptive clause.

Line 122: "Isolates were SNP addresses" I believe this should read "Isolates were [assigned/given] SNP addresses"

Line 288: The three letter abbreviations for the different antimicrobials here are used in supplementary tables, but not defined. Defining these abbreviations here would add clarity. Alternatively, these can be defined in the supplementary materials directly.

Line 348: Suggestion – add a comma at "CipR.MSM5 subclade over time, and geographic" to separate the ideas.

Line 416: Add a comma after "(GenBank accession CP090161)" before reference 19 to offset the descriptive information. This may also be split into two sentences.

Line 464: Remove the comma before citation 80.

Supplementary Table 3: This is not cited in the main text. It is useful information when discussing the fitness costs in the paragraph at line 414 by demonstrating functional expression.

Reviewer #3 (Remarks to the Author):

Thank you for these modifications and for the additional work included in this new version. To the best of my understanding, my previous comments have been addressed in this revision.

Author's Response: Bold

General

We would like to thank the reviewers for their time and helpful comments on this and the previous version of the manuscript. We are pleased that the reviewers feel that we have adequately addressed their previous comments. NB: 'New L' refer to Line numbers in the revised manuscript, 'Old L' refers to the original manuscript submission.

Reviewer #1 (Remarks to the Author):

I have reviewed the updated manuscript and have only minor comments, none pertaining to content. My comments from the first draft have been adequately addressed, and this updated manuscript is suitable for publication. Thanks to the authors for this important work.

Thank you for your kind words and we are pleased that you feel we have adequately addressed your comments made in the first draft of the manuscript.

Minor comments:

Line 65: Citations inside of period.

Thank you for noticing this, the citations have now been moved to the outside of the period.

Line 65/66: A comma after "and further clades and subclades" before citation 14 helps with clarity to offset the descriptive clause.

A comma has now been added as requested.

Line 122: "Isolates were SNP addresses" I believe this should read "Isolates were [assigned/given] SNP addresses"

Thank you, this has been corrected to 'Isolates were assigned SNP addresses'.

Line 288: The three letter abbreviations for the different antimicrobials here are used in supplementary tables, but not defined. Defining these abbreviations here would add clarity. Alternatively, these can be defined in the supplementary materials directly.

We have now added definitions of these abbreviations in both the methods section and to Supplementary Tables 1 and 2.

Line 348: Suggestion – add a comma at "CipR.MSM5 subclade over time, and geographic" to separate the ideas.

A comma has now been added as requested.

Line 416: Add a comma after "(GenBank accession CP090161)" before reference 19 to offset the descriptive information. This may also be split into two sentences.

Thank you, this comma has now been added as requested.

Line 464: Remove the comma before citation 80.

This comma has now been removed before citation 40 (80 in the old manuscript).

Supplementary Table 3: This is not cited in the main text. It is useful information when discussing the fitness costs in the paragraph at line 414 by demonstrating functional expression.

Supplementary Table 2 (3 in the old manuscript) is now cited in the text at New L 207 – 208.

Reviewer #3 (Remarks to the Author):

Thank you for these modifications and for the additional work included in this new version. To the best of my understanding, my previous comments have been addressed in this revision.

Thank you for your time and helpful comments.